# Machine-learning-assisted and real-time-feedback-controlled growth of InAs/GaAs quantum dots

Chao Shen[1,2,3,7], Wenkang Zhan[1,2,7], Kaiyao Xin[2,4], Manyang Li[1,2], Zhenyu Sun[1,2], Hui Cong[2,5], Chi Xu[2,5], Jian Tang[6], Zhaofeng Wu[3], Bo Xu[1,2], Zhongming Wei ![ORCID][2,4], Chunlai Xue[2,5], Chao Zhao ![ORCID][1,2] ✉ & Zhanguo Wang[1,2]

The applications of self-assembled InAs/GaAs quantum dots (QDs) for lasers and single photon sources strongly rely on their density and quality. Establishing the process parameters in molecular beam epitaxy (MBE) for a specific density of QDs is a multidimensional optimization challenge, usually addressed through time-consuming and iterative trial-and-error. Here, we report a real-time feedback control method to realize the growth of QDs with arbitrary density, which is fully automated and intelligent. We develop a machine learning (ML) model named 3D ResNet 50 trained using reflection high-energy electron diffraction (RHEED) videos as input instead of static images and providing real-time feedback on surface morphologies for process control. As a result, we demonstrate that ML from previous growth could predict the post-growth density of QDs, by successfully tuning the QD densities in near-real time from $1.5 \times 10^{10}\,cm^{-2}$ down to $3.8 \times 10^{8}\,cm^{-2}$ or up to $1.4 \times 10^{11}\,cm^{-2}$. Compared to traditional methods, our approach can dramatically expedite the optimization process and improve the reproducibility of MBE. The concepts and methodologies proved feasible in this work are promising to be applied to a variety of material growth processes, which will revolutionize semiconductor manufacturing for optoelectronic and microelectronic industries.

Self-assembled quantum dots (QDs) have attracted great interest due to their applications in various optoelectronic devices[1]. The so-called Stranski-Krastanow (SK) mode in molecular beam epitaxy (MBE) is widely used for growing these high-quality QDs[2]. For specific applications, such as QD lasers, high QD densities are required; while low-density QDs are necessary for other applications, such as single photon sources[1]. However, the outcomes of any QD growth process are a complex function of a large number of variables including the substrate temperature, III/V ratio, and growth rate, etc. Building a comprehensive analytical model that describes complex physical processes occurring during growth is an intractable problem[3]. The optimization of material growth largely depends on the skills and experience of MBE researchers. Time-consuming trial-and-error testing is inevitably required to establish optimal process parameters for the intended material specification. It is needed to build a real-time connection between in situ characterization and material growth status.

[1]Laboratory of Solid State Optoelectronics Information Technology, Institute of Semiconductors, Chinese Academy of Sciences, Beijing 100083, China. [2]College of Materials Science and Opto-Electronic Technology, University of Chinese Academy of Science, Beijing 101804, China. [3]School of Physics Science and Technology, Xinjiang University, Urumqi, Xinjiang 830046, China. [4]State Key Laboratory of Superlattices and Microstructures, Institute of Semiconductors, Chinese Academy of Sciences, Beijing 100083, China. [5]Key Laboratory of Optoelectronic Materials and Devices, Institute of Semiconductors, Chinese Academy of Sciences, Beijing 100083, China. [6]School of Physical and Electronic Engineering, Yancheng Teachers University, Yancheng 224002, China. [7]These authors contributed equally: Chao Shen, Wenkang Zhan. ✉e-mail: zhaochao@semi.ac.cn

Machine learning (ML) is revolutionary due to its exceptional capability for pattern recognition and its potential in approximating the empirical functions of complex systems. It enables researchers to extract valuable insights and identify hidden patterns from large datasets, leading to a better understanding of complex phenomena required to build predictive models, generate new hypotheses, and determine optimal growth conditions for MBE[4–8]. It offers an alternative approach where the growth outcomes for an arbitrary set of parameters can be predicted via a trained neural network, which has been applied to extract film thickness and growth rate information[9,10]. Moreover, by enabling the direct adjustment of parameters during material growth, ML-based in situ control can detect and correct any deviation from expected values in a timely manner[11–13]. Meng et al. have utilized a feedback control to adjust cell temperatures after an amount of film being deposited, thereby capable of regulating $Al_xGa_{1-x}As$ compositions in situ[13]. However, these remain posteriori ML-based approaches, since they require the completion of the growth[10]. Therefore, once the sample characterization results deviate from expectation, it becomes challenging and time-consuming to identify reasons[11].

To fully exploit the effectiveness of real-time connection based on ML, an in situ characterization with continuous and transient working mode is highly desirable to unveil the material status. Reflection high-energy electron diffraction (RHEED) has been widely used to capture a wealth of growth information in situ[14]. However, it still faces the challenges of extracting information from noisy and overlapping images. Traditionally, identifying RHEED patterns during material growth was mainly depended on the experience of growers. Progress has been made in the automatic classification of RHEED patterns through ML that incorporates principal component analysis (PCA) and clustering algorithms[9,14–17]. It surpasses human analysis when a single static RHEED image is gathered with the substrate held at a fixed angle[16,18]. However, this may result in an incomplete utilization of the temporal information available in RHEED videos taken with the substrate rotating continuously[16,19,20]. Although few researchers have highlighted the importance of analyzing RHEED videos, their ML models were still based on PCA and encoder-decoder architecture with a single image input[16,21]. To date, it is still hard to build a real-time connection between in situ characterization and material growth status, which is a prospective way to realize the precise control on material specifications.

In this work, we proposed a real-time feedback control based on ML to connect in situ RHEED videos and QD density, automatically and intelligently achieving a low-cost, efficient, reliable, and reproduceable growth of QDs with required density. By investigating the temporal evolution of RHEED information during the growth of InAs QDs on GaAs substrates, the real-time status of the sample was better reflected. We applied two structurally identical 3D ResNet 50 models in ML deployment, which takes three-dimensional variables as inputs to determine the QD formation and classify density[22,23]. We showed that ML can simulate and predict the post-growth material specifications, and based on which, growth parameters are continuously monitored and adjusted. As a result, the QD densities were successfully tuned from $1.5 \times 10^{10}\,cm^{-2}$ down to $3.8 \times 10^{8}\,cm^{-2}$, and up to $1.4 \times 10^{11}\,cm^{-2}$, respectively. The methodology can also be adapted to other QD systems, such as the droplet QDs, the local droplet etching, and manhole infilling QDs with distinct RHEED features[24–27]. The effectiveness of our approach by taking full advantages of in situ characterization marks a significant achievement of establishing a precise growth control scheme, with the capabilities and potentials of being extended to large-scale material growth, reducing the impact on material growth due to instability and uncertainty of MBE operation, shortening the parameter optimization cycle, and improving the final yield of material growth.

## Results

### Sample structure and data labeling
Our method relied on an existing database of growth parameters with corresponding QD characteristics, which helped us decide the growth parameters needed to achieve a target density for QDs. The sample structure is shown in Fig. 1a, with density results of 30 QD samples summarized in Table 1, which are correlated with the RHEED videos. Each QD growth was repeated on average 4 times, leading to the generation of 120 RHEED videos. The samples grown without QDs have flat surfaces determined by an atomic force microscope (AFM) as shown in Fig. 1b, corresponding to streaky RHEED patterns in Fig. 1f, which was artificially labeled with "zero" referring to the density of QDs on the surface[27]. With the increase of QD density, it was found that the RHEED gradually converted from streaky to spotty patterns. The QDs density of ~$2.9 \times 10^{9}\,cm^{-2}$ from Fig. 1c corresponds to the RHEED pattern consisted of streaks and spots in Fig. 1g[27]. As shown in Fig. 1d, h, with the further increase of QDs density reaching $1.9 \times 10^{10}\,cm^{-2}$, the arrangement of RHEED pattern showed typical spot features[28–30]. Upon conducting a more thorough analysis of the RHEED patterns, we identified the patterns transition from having both streaks and spots to only spots, corresponding to a density of $1 \times 10^{10}\,cm^{-2}$ in AFM images. So, we assigned the "low" label to densities below $1 \times 10^{10}\,cm^{-2}$, the "middle" label to densities ranging from $1 \times 10^{10}\,cm^{-2}$ to $4 \times 10^{10}\,cm^{-2}$. As the density exceeds $4 \times 10^{10}\,cm^{-2}$, the spot features become more rounded and show higher brightness, we labeled densities exceeding $4 \times 10^{10}\,cm^{-2}$ as "high" (see Supplementary Information for RHEED characteristics of QDs with different labels, S1)[31–33]. Fig. 1e, i show the typical AFM and RHEED images after the QD formation, respectively, with the QD density of ~$1.0 \times 10^{11}\,cm^{-2}$. It is worth noting that we keep the labels to the RHEED videos the same before and after the QD formation. The RHEED information before the QD formation can be used to determine the resulting QD density grown under the same conditions, enabling us to adjust the material growth parameters before the QD formation.

### Program framework and video processing
The design framework of our program is illustrated in Fig. 2a. An appropriate scheme was designed to pre-process the RHEED video data. Then, a ML model was selected based on the pre-processing results. Furthermore, we also determined the way to adjust parameters according to output results of the model.

RHEED videos taken during growth were first deconstructed into multiple temporal images and utilized as input for our model, which is a technique for breaking down video processing, offering a more versatile and efficient approach to analyze and process information in dynamic scenes[34]. The original image collected has uncompressed 4 channels of 8-bit depth color with a resolution of $1920 \times 1200$. The software cropped a square area from the image, as shown in Fig. 2b, immediately compressed to $300 \times 300$ by zero-order sampling and converted it to a single $4 \times 300 \times 300$ matrix (see Supplementary Information for the cropping area of RHEED images, S2). To efficiently utilize the temporal information within RHEED data, we used the latest RHEED image as a starting point, acquired an additional several consecutive images before, and bundled as the original RHEED dataset for the model, as shown in Fig. 2c[35]. Subsequently, each image was converted to a single-channel 8-bit grayscale image based on the luminance information[16,21], and stitched into an $N \times 300 \times 300$ 3D matrix through shift registers, as shown in Fig. 2d. Additionally, we modified the convolutional filtering order. Traditional Convolutional Neural Networks (CNNs) tend to uniformly sample color channels, which is effective for images with rich color information, but not suitable for our data[36,37]. To address this issue, we designed an image processing approach that uses longitudinal size information instead of color channel information, which can improve the correlation between

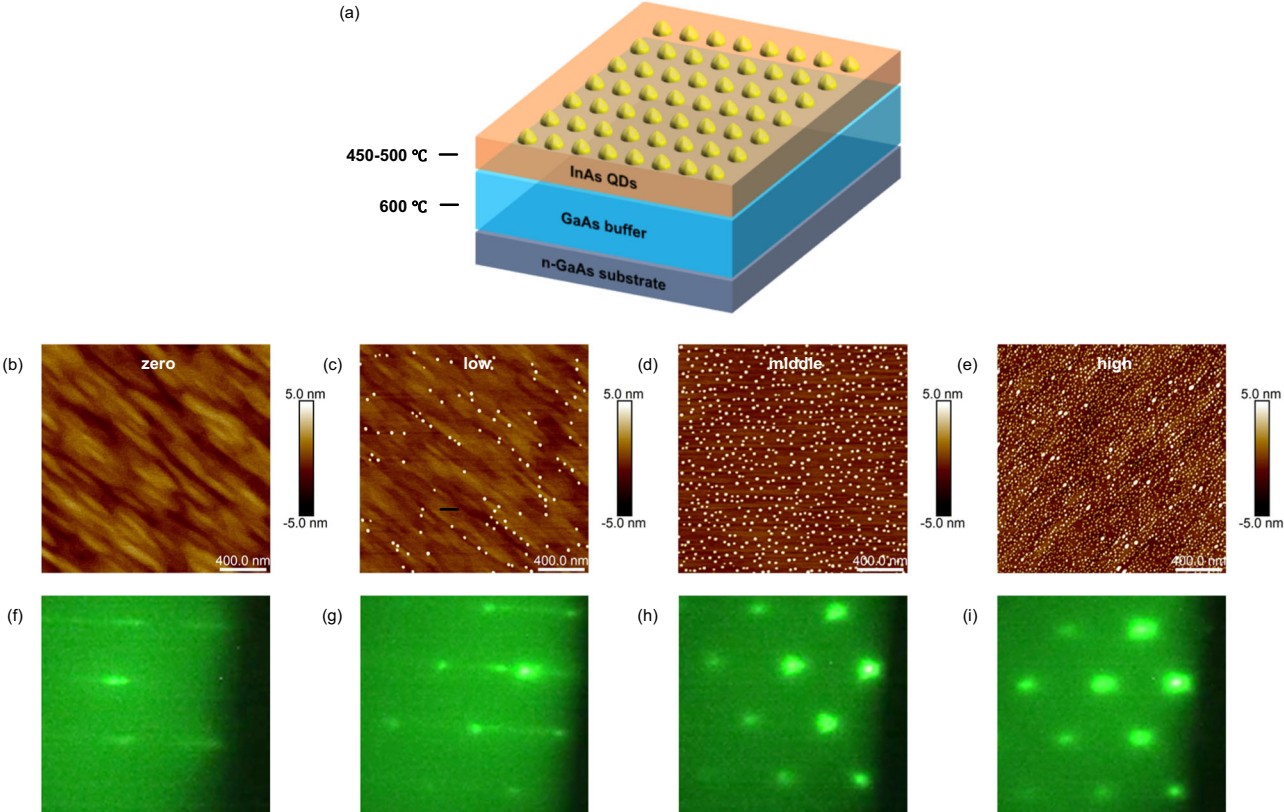

**Fig. 1 | Quantum dot (QD) sample structure and data labeling. a** The schematic of the sample structure. The thick GaAs buffer layer is grown on an n-type GaAs substrate at 600 °C. Subsequently, the substrate is cooled to 450–500 °C for the growth of InAs QDs. **b**–**e** The 2 μm × 2 μm atomic force microscope (AFM) images of QDs with varied labels. "zero": there are no QDs on the surface. "low": the InAs QDs sample with density less than $1 \times 10^{10}$ cm$^{-2}$. "middle": the InAs QDs sample with density range from $1 \times 10^{10}$ cm$^{-2}$ to $4 \times 10^{10}$ cm$^{-2}$. "high": the InAs QDs sample with density higher than $4 \times 10^{10}$ cm$^{-2}$. **f**–**i** Corresponding reflection high-energy electron diffraction (RHEED) images with varied labels.

channels, reduce unnecessary calculations, and improve data processing efficiency[38–40]. Therefore, the size of the preprocessed data was $N \times 300 \times 1 \times 300$, representing the image number, image width, image color channel, and image length. We then compared the impact of choosing different numbers of images on model training accuracy and speed, ultimately determined that selecting 8 images was the appropriate choice (see Supplementary Information for the number of images selected and speed management, S3 and S4). Finally, we conducted preprocessing for all data before model training to reduce the time. We have also applied data augmentation techniques to the training data, including adjusting image curves, cropping images, and scaling images, which effectively increase data diversity and improve the model's generalization ability. This process resulted in the acquisition of ~360,000 NumPy arrays with each sample ~704 KB in size.

## Model construction and evaluation

The ResNet 50 model has emerged as our preferred choice for several reasons, including its capability to address gradient vanishing issues and automatically learn methods for extracting features from raw data without manual feature engineering or dimensionality reduction, rapid convergence, and suitability for deep training. When processing video frame data, 3D convolution proves more effective in extracting spatial and temporal dimension information from videos than the 2D convolution used in the standard ResNet model. The difference between our network and the original ResNet lies in the dimensionality of the convolutional kernel and batch normalization operations[23]. Our 3D ResNet 50 model employs 3D convolutions and 3D batch normalization. This spatiotemporal-feature-learning method enables the 3D ResNet 50 model to better capture the dynamic information in videos

compared to 2D CNN, resulting in an enhanced ability to identify and differentiate the categories of videos[41,42]. Furthermore, we have adjusted the structure of identity shortcuts to reduce information loss during downsampling and reduced the frequency at which the model doubles the number of channels after passing through the residual structure (see Supplementary Information for the detailed structure of the ResNet 50 model and the principle of the basic residual block in ResNet, S5 and S6). We also compared the ResNet 50 model and other models, and the results revealed that ResNet 50 outperforms the other models significantly (see Supplementary Information for comparison of training and validation results of different models, S7)[23].

We trained models for judging the QDs formation and classifying density, which are called the "QDs model" and the "density model", respectively (see Supplementary Information for model development and training result, S8). As shown in Fig. 3a, the output of the QDs model consists of just two categories, represented as "Yes" or "No", indicating whether the QD has formed. In contrast, the output of the density model consists of "zero", "low", "middle", and "high" labels, corresponding to different QD density prediction results.

The accuracy and loss of the QDs and density models during the training are shown in Fig. 3b, c. Overall, the improvement trend of accuracy for the two models is evident. As the training of the two models progresses, the Acc curve exhibits an overall upward trend, while the Loss curve shows a downward trend. Due to the more significant changes in loss values compared to accuracy values, additional simple polynomial fitting and the first derivative of the Loss curve were performed to obtain Trend and Change curves. It can be observed that the Train Trend and Train Change show a stable trend close to zero throughout the entire training process. Furthermore, the change in

**Table 1 | QD densities and labels assigned for the samples in the dataset, the density of QDs was determined by counting the number of QDs in the AFM image**

| Sample | Density ($cm^{-2}$) | Label |
|---|---|---|
| 1 | 0.000E + 00 | zero |
| 2 | 0.000E + 00 | zero |
| 3 | 0.000E + 00 | zero |
| 4 | 3.960E + 08 | low |
| 5 | 6.936E + 08 | low |
| 6 | 1.700E + 09 | low |
| 7 | 3.900E + 09 | low |
| 8 | 9.000E + 09 | low |
| 9 | 9.442E + 09 | low |
| 10 | 1.250E + 10 | middle |
| 11 | 2.035E + 10 | middle |
| 12 | 2.416E + 10 | middle |
| 13 | 2.616E + 10 | middle |
| 14 | 2.666E + 10 | middle |
| 15 | 2.768E + 10 | middle |
| 16 | 2.920E + 10 | middle |
| 17 | 2.999E + 10 | middle |
| 18 | 3.266E + 10 | middle |
| 19 | 3.330E + 10 | middle |
| 20 | 3.389E + 10 | middle |
| 21 | 3.537E + 10 | middle |
| 22 | 3.629E + 10 | middle |
| 23 | 3.725E + 10 | middle |
| 24 | 3.836E + 10 | middle |
| 25 | 4.740E + 10 | high |
| 26 | 6.480E + 10 | high |
| 27 | 7.040E + 10 | high |
| 28 | 9.493E + 10 | high |
| 29 | 1.177E + 11 | high |
| 30 | 1.691E + 11 | high |

Valid Loss can better reflect the model's performance compared to Train Loss. The Valid Trend and Valid Change curves fluctuate significantly and gradually stabilize, indicating that the model is progressively learning information from initial states and achieving stable convergence, resulting in a minor deviation between the model output and the dataset. Throughout the model training process, the persistent fluctuations and plateaus in the "QDs model" and "density model" accuracy were attributed to the data diversity and the manual label assignment process instead of limitations inherent in the model itself. After 120 training cycles, the "QDs model" maintained a validation accuracy of over 90%, as shown in Fig. 3b, ultimately achieving a final average validation accuracy of 94.4%. The fluctuations observed in the "QDs model" validation process predominantly originate from a specific time before the QD formation when the RHEED streaks still dominated most of the collected images. As the QDs are about to form, the spot features closely associated with QD formation are often faint and overshadowed by the streak features. As depicted in Fig. 3c, the validation accuracy of the "density model" continues to exhibit fluctuations even after 100 epochs, primarily attributed to label overlap when low-density QD data exhibit an increase in density with deposition[43,44]. However, it consistently maintained a validation accuracy of over 80%, and the final average validation accuracy reached 95.1%.

Furthermore, we collect multiple results from the QDs and density models outputs when deploying. The result with the highest probability is selected to determine whether the adjustment to the growth parameters is necessary. This approach helps mitigate the impact of randomness and variability, reducing the likelihood of operational errors by the model during software deployment[45–47]. Moreover, we converted the model into the open standard Open Neural Network Exchange (ONNX) format. When deployed in a LabVIEW environment equipped with TensorRT accelerators, the QDs and density models can each produce results at an approximate rate of 1 sample per second, effectively meeting the application's requirements (see Supplementary Information for hardware wiring scheme for model deployment and Program interface and deployment environment, S9 and S10). Additionally, the model is universally applicable to other systems with different wobble characteristics without a new training cycle (see Supplementary Information for the QDs and density models with different wobble characteristics, S11).

To enable early intervention in the growth process before the QDs formation, we developed a control logic for the LabVIEW program, as shown in Fig. 3d. After opening the shutter, the substrate temperature remains unchanged if the label output from the density model aligns with the preset target before QD formation. If the output label's density from the density model exceeds the target density, the substrate temperature increases; otherwise, it decreases. Once the QDs model recognizes QD formation, the shuttle will remain open until the density model's output matches the target. At this point, it closes the shutter to complete the growth process. This control logic adjusts the substrate temperature continuously, gradually increasing the likelihood of growing QDs with the target density.

## Controlled growth of low-density QDs

We have grown a reference sample with QD density of $1.5 \times 10^{10}\,cm^{-2}$, corresponding to label "middle" (see Supplementary Information for details of the reference sample, S12). Subsequently, we set the initial substrate temperature equivalent to that used in the reference sample and conducted in situ control experiment with the "low" or "high" label as the target. After growth, we compiled each frame of the RHEED video captured during the growth into a sequence and analyzed. In order to distinguish different growth stages under model control, we marked the QD formation time and the In shutter closing time judged by the model with blue and yellow lines in the Figs. 4, 5. In addition, since the "zero" labels appear rarely and have little impact on output results, it was not included in the figures.

As shown in Fig. 4a, the substrate temperature increased by 44 °C from the beginning of the sequence until the In shutter was closed (see Supplementary Video 1 for the experiment). This indicates that the initial substrate temperature was not suitable for the "low" label. In addition, the blue line and yellow line are in the same sequence, indicating that when the QDs are formed, the "density model" has already determined that the RHEED sequences are consistent with the "low" label. As shown in Fig. 4b, the RHEED images mainly show streaks in the initial growth stage of QDs; it kept even before the QD formation, as shown in Fig. 4c. This is attributed to the relatively flat surface of the sample in the early stage of QD growth. After the QD formation, a pattern coexisting streaks and spots was immediately observed as shown in Fig. 4d, consistent with the "low" label characteristics[27]. The QDs has a density of $3.8 \times 10^8\,cm^{-2}$, with an average diameter of 50.5 nm and height of 10.0 nm, as shown in the AFM image in Fig. 4e.

Figure 4f shows the "QDs model" results of the experiment with the "low" label as the target. To effectively depict data trends, we have incorporated running average plots to illustrate the evolution of the model's output results. We classified RHEED videos as "Yes" or "No" based on observations of QDs formation. At the beginning of growth, the primary output of the QDs model was "No" from the initial to the 200th sequence, and the probability of outputting "Yes" was only 7%. Even though the proportion of outputting "Yes" increases to 15%, the likelihood of the model outputting "No" remains higher from the

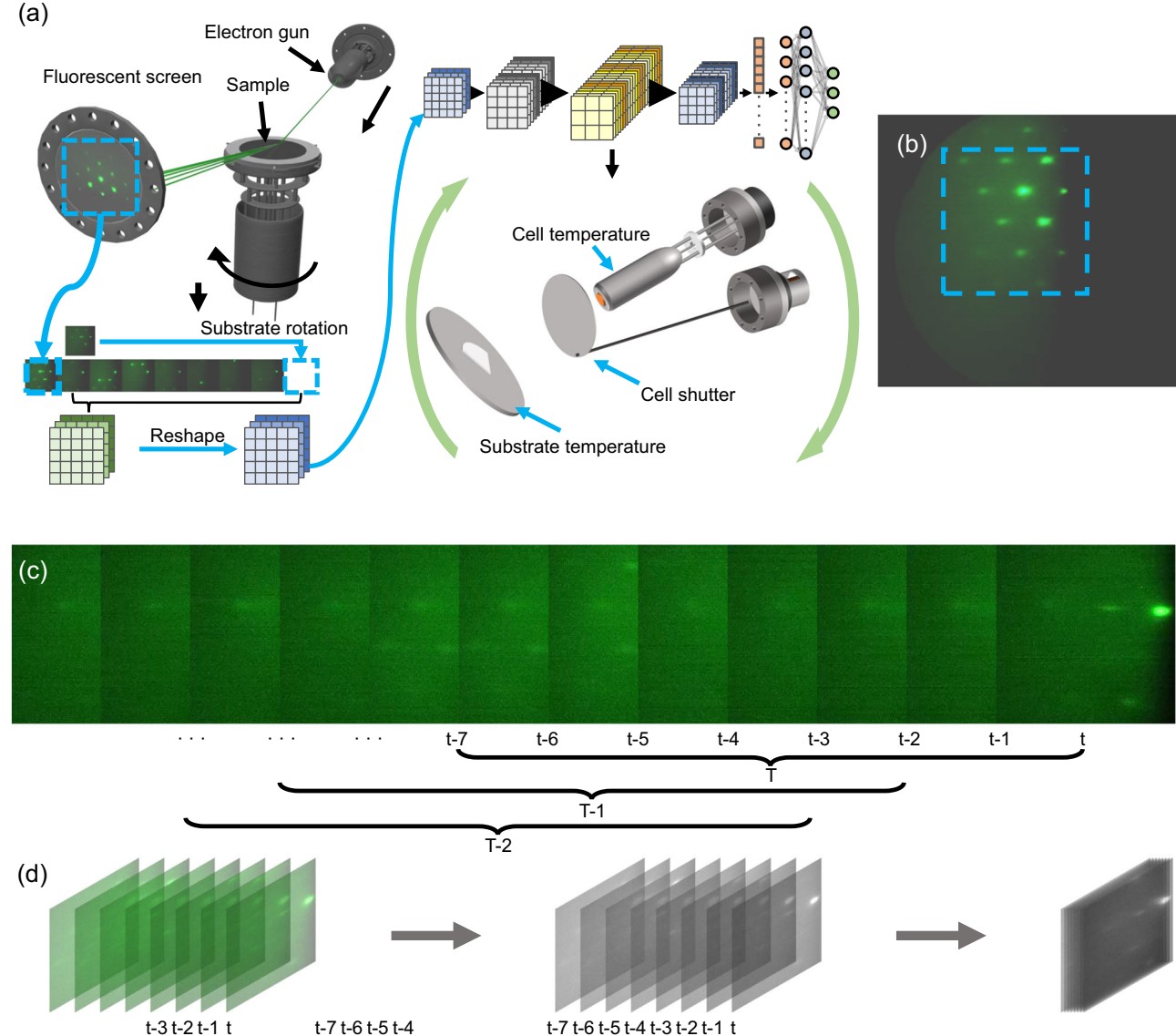

**Fig. 2 | Control program framework and video processing principle. a** The framework of the program. An electron beam generated by an electron gun is continuously applied to the rotating substrate, creating a diffraction image frame by frame on the fluorescent screen. Multiple images are then converted to a new matrix, and the reshaped matrix is then transferred to the model for generation of results to guide the adjustment of material growth parameters. **b** A typical reflection high-energy electron diffraction (RHEED) image taken from the camera and the cropping area. The area marked by the blue square is an effective selection area that the software must handle and subsequently provide as input to the model. **c** Continuous sampling method for RHEED images. The software processes images with data at position T during each iteration. T is comprised of several sub-images taken sequentially from time t, including t-1, t-2, and so forth. Similarly, the data positions processed by the software in the previous iteration are T-1, T-2, and so on. **d** Processing method for sampled images. The data at a specific position T is initially acquired. Subsequently, each data point within T, denoted as t, t-1, t-2, and so forth, is transformed into a 2D matrix containing only black and white channels. Finally, these individual 2D matrices are stitched into a 3D matrix.

200th to the final 40 sequence before the blue line. When the growth process approaches the QDs formation, we observed a significant increase from 15% to over 27% of the "Yes" label. This trend continues until the shutter closes, with over 55% probability of outputting "Yes". This indicates that the QDs model exhibits a certain sensitivity to the presence of low-density QDs (see Supplementary Information for another controlled growth experiment of low-density QDs, S13).

Figure 4g shows the "density model" results of the experiment with the "low" label as the target. From the initial to the 200th sequence, the probability of output "low" is significantly lower than that of combined "middle" and "high". However, with the real-time feedback control of growth conditions, it is evident that the probability of output "low" approaches over 50%. The growth conditions have been gradually adjusted to better suit the growth of low-density

QDs. It becomes apparent from the sequence before and after the blue line that the probability of the model outputting "low" surpasses 55% and persists until the end of the growth.

## Controlled growth of high-density QDs
We also conducted an in situ control experiment with the "high" label as the target (see Supplementary Video 2 for the experiment). From Fig. 5a, it can be observed that the substrate temperature decreased by a total of 24 °C, indicating that the initial substrate temperature was not suitable for "high" label. In addition, the blue line did not overlap with the yellow line, which indicates the RHEED image with "high" label did not appear during the initial stage of the QD formation. In the initial stage of InAs growth, the RHEED pattern mainly showed streaky features, as shown in Fig. 5b. As the deposition amount increased, the streaks

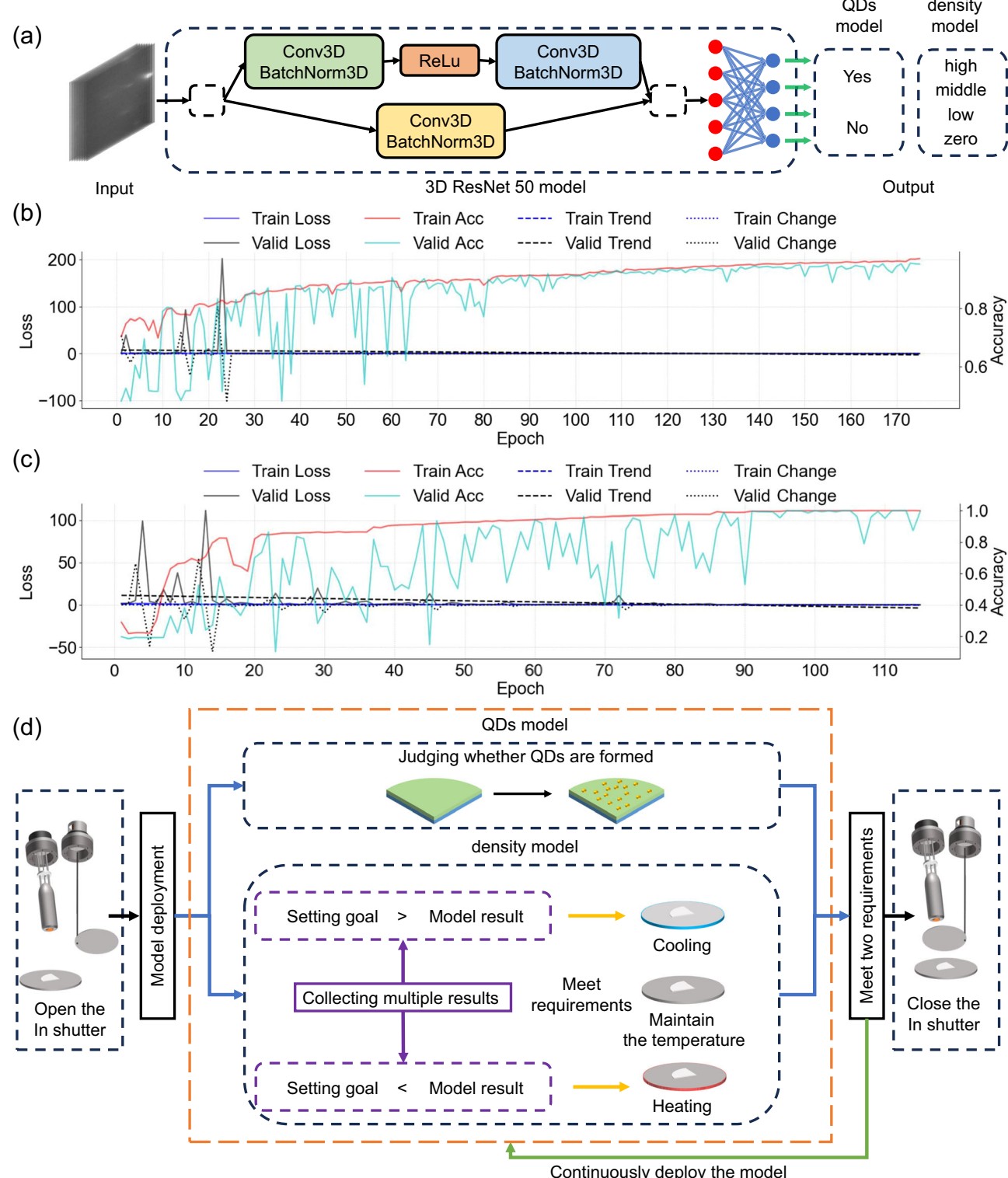

**Fig. 3 | Model construction, evaluation, and deployment. a** A simplified architectural diagram of the model. The preprocessed data is processed by a 3D ResNet 50 model, which comprises both residual and fully connected structures and outputs classification results. Conv3D is a 3D convolution layer. BatchNorm3D is a 3D batch normalization layer. ReLU represents the rectified linear unit activation function. Training performance of (**b**) "QDs model", and (**c**) "density model". Train: training. Valid: validation. Acc: accuracy. Loss: loss. The Loss and Acc curves are derived from results during model training. The Trend and Change curves are simple polynomial fit and the first derivative of the Loss curve, respectively. (**d**) A

control logic diagram for the model deployment. After opening the shutter, the substrate temperature remains unchanged if the label output from the density model aligns with the preset target before quantum dot (QD) formation. If the output from the density model exceeds the target density, the substrate temperature increases; otherwise, it decreases. Once the QDs model recognizes QD formation, the shuttle will remain open until the density model's output matches the target. At this point, it closes the shutter to complete the growth process. Source data are provided as a Source Data file.

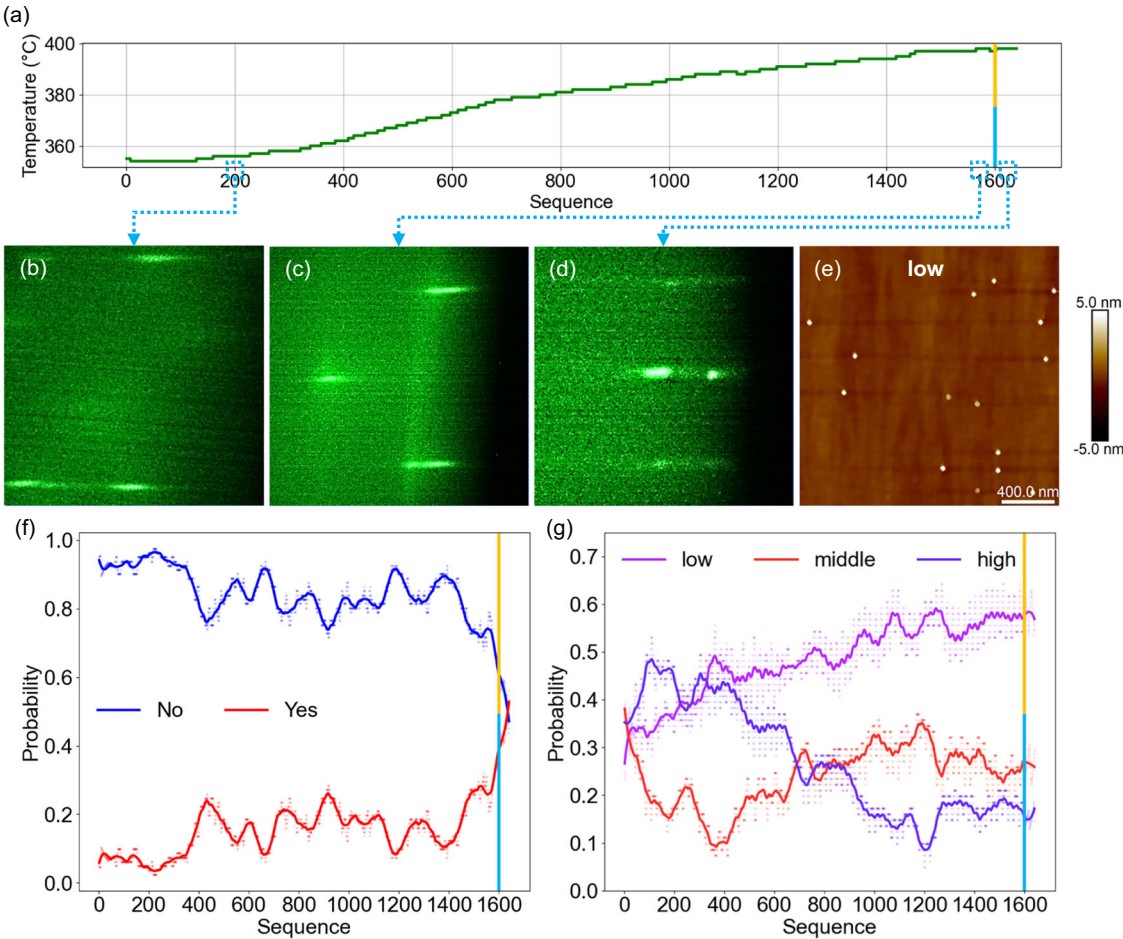

**Fig. 4 | Controlled growth process of low-density quantum dots (QDs).**
Experiment with the "low" label as the target. Blue lines: the QD formation time.
Yellow lines: the In shutter closing time. **a** Substrate temperature during growth.
The reflection high-energy electron diffraction (RHEED) image (**b**) captured at
200th frame after growth; (**c**) before the QD formation; (**d**) after the QD formation;
(**e**) the 2 μm × 2 μm atomic force microscope (AFM) image of the sample; the pre-
diction results of (**f**) the "QDs model" and (**g**) the "density model". Dots: probability
statistics of different labels based on the model results. Lines: running average plots
of dots. Source data are provided as a Source Data file.

gradually became blurred before the QD formation, indicating that a
certain amount of InAs has already deposited and caused the substrate
surface to become rough, as shown in Fig. 5c. The RHEED pattern in
Fig. 5d showed streaks and spot feature after the QD formation like
those shown in Fig. 4d, which are typical features of the "low" label. This
explains why the blue line and the yellow line in Fig. 5a did not coincide.
As the In shutter closed, and the RHEED pattern exhibited spots, as
shown in Fig. 5e. The QDs density is ~1.4 × 10¹¹ cm⁻² with an average
diameter of 33.5 nm and height of 3.2 nm, as shown in Fig. 5f.

In high-density growth experiments, the shutter remains open after
the formation of QDs for a period. As depicted in Fig. 5g, the QDs model
predominantly outputs "No" before QDs formation, with the probability
of outputting "Yes" reaching a maximum of 30%, as evidenced from the
initial to around 1000th sequence. As we approach the formation of
QDs, the probability of the QDs model outputting "Yes" approaches
100% around the blue line, indicating high accuracy in recognizing spot
patterns after the QD formation. It is noteworthy that there is still a
distance between the last "No" label and the blue line, since we confirm
the QD formation by collecting multiple results of the model.

The output results of the density model are also presented in
Fig. 5h. Before QDs formation, the density model rarely outputs "high".
However, as the growth process approaches the stage of QD forma-
tion, the probability of "high" label output slowly increases to 15%
around the blue line. Following the formation of QDs, the probability
of the density model outputting "high" rapidly increases to over 30%

around the 1100ᵗʰ sequence. Due to the unique nature of QD systems,
closing the shutter at this point would restrict the formation of high-
density QDs[48,49]. As the growth process continues and approaches the
moment the shutter is closed, it becomes evident that the probability
of the model outputting "high" rapidly increases to over 70%. This also
indicates the successful growth of high-density QDs in our experiment.

## Discussion
Furthermore, we also set the initial conditions favorable for the growth
of high-density QDs and conducted experiments with the "low" label as
the target (see Supplementary Information for controlled growth of
low-density QDs with initial high-density growth conditions, S14). The
densities of QDs we demonstrated in the experiments are typical for
single photon sources and QD lasers. Note our experiments have a
certain failure rate due to data labeling, the accuracy of the model, and
limited dataset[50–52]. In our experiment, each prediction takes a few
seconds, which is limited by the graphics card of our setup; the com-
munication between the server and MBE controller takes seconds
which is determined by wiring, the baud rates of serial communication,
and the output capacity of the controller. By upgrading the hardware
of our MBE system, a few tens of milliseconds of response can be
achieved using the same workflow developed in this report. Moreover,
as the breakthroughs in data label partitioning methods and dataset
expansion in the future, we anticipate achieving finer-grained density
regulation during the material growth process.

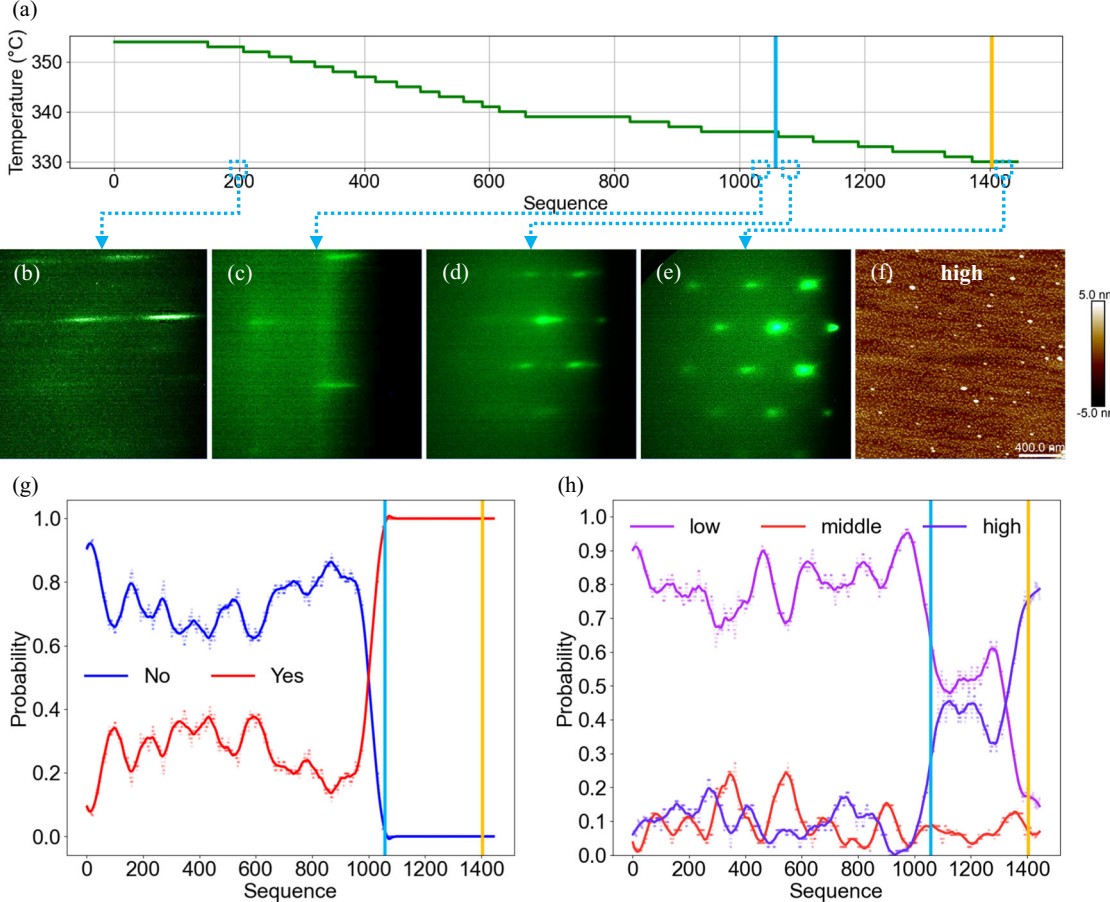

**Fig. 5 | Controlled growth process of high-density quantum dots (QDs).**
Experiment with the "high" label as the target. Blue lines: the QD formation time. Yellow lines: the In shutter closing time. **a** Substrate temperature during growth. The reflection high-energy electron diffraction (RHEED) image (**b**) captured at 200th frame after growth; **c** before the QD formation; **d** after the QD formation; **e** before the end of the growth; **f** the 2 µm × 2 µm atomic force microscope (AFM) image of the sample; the prediction results of (**g**) the "QDs model" and (**h**) the "density model". Dots: probability statistics of different labels based on the model results. Lines: running average plots of dots. Source data are provided as a Source Data file.

In conclusion, we developed a metrology to control the material's properties grown by MBE in situ. Specifically, we showed that a neural network could accurately predict the post-growth density of QDs within a wide range by utilizing RHEED videos information of as-grown samples. We applied the method to InAs/GaAs QDs by tuning growth parameters in near real-time and validated it to in situ control the QD density, which was usually only possible by trial-and-error. The capability can significantly reduce the time and the number of experimental iterations required for development and optimization to achieve material specifications. In this proof-of-concept study, we trained a network to simulate a specific type of MBE growth task by varying the substrate temperature. In practice, one would train the network to the tasks with a full range of system metadata. In this way, the network would accrue an "understanding" of the complex relationships among the numerous samples and system parameters that affect the growth outcomes. Our investigations highlighted the considerable strength and practicality of the growth metrology to achieve high-quality epitaxial thin films for various applications. Moving forward, there is a huge potential for such networks to be deployed for defect detection, identification, and repair during material growth at an early stage.

## Methods
### Material growth
The InAs QD samples were grown on GaAs substrates in the Riber 32 P MBE system, equipped with arsenic (As) valved cracker, indium (In), and gallium (Ga) effusion cells. $As_4$ was used in the growth by keeping the cracker temperature at 600 °C. The beam equivalent pressure (BEP) was used to evaluate the flux of the III and V elements and calibrate their ratio[53]. A C-type thermocouple measured the substrate temperature, and the growth rates were calibrated through the RHEED oscillations of extra layers grown on GaAs substrates. The substrate was outgassed at 350 °C in the buffer chamber, then heated to 620 °C in the growth chamber for deoxidation. The substrate was then cooled to 600 °C for the growth of the GaAs buffer layer. The sample structure is shown in Fig. 1a. The BEP of In was $1.4–1.5 × 10^{-8}$ Torr, while the BEP of As was $0.95–1.0 × 10^{-7}$ Torr. The QDs were formed in the SK growth mode after depositing 1.7 ML of InAs at a temperature of 450–500 °C and a rate of 0.01 ML/s. The initial growth showed a relatively flat planar structure, pertaining to the wetting layer growth[54–59]. The formation of InAs QDs was verified by the streaky-spotty RHEED pattern transition[60].

### Material characterization
RHEED in the MBE growth chamber enabled us to analyze and monitor the surface of the epilayer during the growth. RHEED patterns were recorded at an electron energy of 12 kV (RHEED 12, from STAIB Instruments). A darkroom equipped with a camera was placed outside the chamber to continuously collect RHEED videos with the substrate rotating at 20 revolutions per minute. The exposure time was 100 ms, with a sampling rate of 8 frames per second. To achieve a clear correlation of RHEED with QD density, we also characterized the surface morphology of the InAs QDs using AFM (Dimension Icon, from Burker).

## Hardware wiring scheme

Our model is deployed on a Windows 10 computer system with an AMD R9 7950X CPU, 64GB of memory, an NVIDIA 3090 graphics card, and a 2TB solid-state drive. The program primarily obtains information from the temperature controllers, shutter controllers, and the camera (Fig. S11). We use a USB 3.0 interface to connect the camera to the computer. Temperature controllers are connected in series using the Modbus protocol to simplify the wiring. To reduce the delay in substrate temperature control, our program solely looks up information by address from the temperature controllers connected to the substrate.

## Model construction environment

The model development occurred in the Jupyter Notebook environment on the Ubuntu system, utilizing Python version 3.9. We installed PyTorch based on CUDA 11.8 in this environment to leverage graphics card operations. The original video data was rapidly converted into the NumPy arrays format using code with random data augmentation within the Ubuntu system. Subsequently, the processed data was stored on our computer's hard disk.

## Program interface and deployment environment

The program was developed using LabVIEW, the built-in NI VISA, NI VISION, and Python libraries for data acquisition and processing (see Supplementary Information for program interface and deployment environment, S12). The program also employs ONNX for model deployment and TensorRT for inference acceleration and allows users to set targets for the desired QD density. Once the substrate and In cell temperatures stabilize, the program can be initiated. Firstly, the program checks the shutter status and controls the substrate and cell temperatures. Once the In shutter is open, the model starts to analyze RHEED data in real-time. The model outputs are displayed as numerical values at the top of the interface and are converted to corresponding label characters on the right side. The growth status can be displayed in the "Reminder Information". Before and after the QDs formation, the "Reminder Information" will show up as "Stage 1" and "Stage 2", respectively. It will display "Finished" when the model results meet the targets several times after the QD formation.

## Reporting summary

Further information on research design is available in the Nature Portfolio Reporting Summary linked to this article.

## Data availability

The datasets generated during and/or analyzed during the current study are available in the Figshare repository, https://doi.org/10.6084/m9.figshare.24347053[61]. Source data are provided with this paper.

## Code availability

The codes supporting the findings of this study are available from the corresponding authors upon request.

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

## Acknowledgements

This work was supported by the National Key R&D Program of China (Grant No. 2021YFB2206503, C.Z.), National Natural Science Foundation of China (Grant No. 62274159, C.Z.), the "Strategic Priority Research Program" of the Chinese Academy of Sciences (Grant No. XDB43010102, C.Z.), and CAS Project for Young Scientists in Basic Research (Grant No. YSBR-056, C.Z.).

## Author contributions

C.S. and W.K.Z. contributed equally. C.Z. conceived of the idea, designed the investigations and the growth experiments. C.S., W.K.Z. and M.Y.L. performed the molecular beam epitaxial growth. C.S., K.Y.X., H.C. and C.X. did the sample characterization. C.S., C.Z., Z.Y.S., J.T., Z.F.W, Z.M.W. and C.L.X. wrote the manuscript. C.Z. led the molecular beam epitaxy program. B.X. and Z.G.W. supervised the team. All authors have read, contributed to, and approved the final version of the manuscript.

## Competing interests

The authors declare no competing interests.

## Additional information

**Supplementary information** Details on RHEED characteristics of QDs with different labels, the cropping area for RHEED images, the number of images selected, speed management, the detailed structure of the ResNet 50 model, the principle of the basic residual block in ResNet, comparison of training and validation results of different models, model development and training results, hardware wiring scheme for model deployment, program interface and deployment environment, the QDs and density models with different wobble characteristics, details of the

reference sample, another controlled growth experiment of low-density QDs, controlled growth of low-density QDs with initial high-density growth conditions. The video demonstrated QD growth experiments with "low" and "high" labels as targets, respectively. The online version contains supplementary material available at https://doi.org/10.1038/s41467-024-47087-w.

