## [Peer Review File · Nature Communications]

Machine-Learning-Assisted and Real-Time-Feedback-Controlled Growth of InAs/GaAs Quantum DotsREVIEWER COMMENTS

Reviewer #1 (Remarks to the Author):

As someone who started the research career as a MBE grower, I still remember the nightmare experience of trying to control the quantum dot (QD) density during the MBE growth process. Therefore I am very excited to see the results from the authors. They developed a machine learning model for training RHEED videos instead of static images, which provides the long sought-after real-time feedback/control on surface morphologies. I would like to emphasize that the control on the quantum dot density is indispensable for many important applications. For example, the QD lasers require high density while the single photon sources require low density. The authors demonstrated a QD density tuning of over several orders of magnitude. The experimental data are well and clearly presented and the machine learning model is described. The supplementary materials are also nicely prepared. I believe that this result will be highly welcome by the MBE growth community. Therefore I highly recommend the publication as it is. Maybe a minor remark: the authors can comment how their methodology can be adapted to other QD systems (other than the SK QDs), for example, the droplet QDs, the local droplet etching and manhole infilling QDs.

Reviewer #2 (Remarks to the Author):

The report details the use of machine learning for tuning the quantum dot density of MBE grown materials based off of RHEED videos. The use of video instead of still images and the implementation during substrate rotation are challenging and notable accomplishments of the present report. I find this report to be a compelling use of machine learning applied to epitaxy worthy of publication after minor clarifications mostly directed to the more general audience not familiar with machine learning. 1) More detail should be discussed about the Resnet algorithms used. As written the unfamiliar reader might interpret the authors as originating this structure. Instead, the authors merely applied this structure to show how it can be used to learn and predict QD density. The authors should clarify any specific changes relative to the standard Resnet approach and make clear any differences they innovated or if they merely used the pre-existing image processing approach. 2) Some discussion of data size and throughput (speed) management should be included. Having done some of these calculations, I am astonished that a 8x300x300 pixel image frame video can be processed real time without data reductions like found in PCA for example. I am even more astonished this can be done within the notoriously slow LabVIEW environment. This discussion should include the computational difficulty and speed of the training process and separately, the real time inference. These limitations are the key to practical implementation. Finally, it is well known that each MBE system has physical limitations in the wobble of substrates. Each tool is different. The authors should address the universal applicability of the developed model. Can their model be applied to other systems with different wobble characteristics or will a new training cycle be required for every tool. In general, I find this method more useful for production MBE than research owing to the repetitive nature of production versus the day-to-day variance of research activities. nevertheless, it is an impressive display of ML power.

Reviewer #3 (Remarks to the Author):

The paper highlights the growth of InAs/GaAs quantum dots (QDs) using molecular beam epitaxy (MBE) method. The RHEED patterns during QDs formation were analyzed using the 3D ResNet 50 machine learning model. This model interprets RHEED videos in real-time to offer insights into QD density during growth. Using this feedback, the system adjusts substrate temperature automatically to control QD growth with specific density labels (zero, low, middle, high). Two models, the "QDs model" and "density model", were trained to validate QD formation and measure density. Utilizing these models, the authors successfully synthesized QDs with targeted densities: a "low" label at $3.8 \times 10^8 \text{ cm}^{-2}$ and a "high" label at $1.4 \times 10^{11} \text{ cm}^{-2}$.

Numerous studies have explored the regulation of InAs quantum dot (QD) growth density through the MBE experiments, as evidenced by works such as <https://doi.org/10.1063/1.1327613>.

Additionally, scholars have investigated feedback control in MBE experiments ([https://doi.org/10.1016/S0952-1976\(98\)00024-4](https://doi.org/10.1016/S0952-1976(98)00024-4)) and alternative optimization models for handling reflection high-energy electron diffraction (RHEED) video in film growth processes (<https://doi.org/10.1186/s40580-023-00359-5>). In this study, a novel approach utilizing the 3D ResNet 50 model (originating from the ResNet model: <https://doi.org/10.48550/arXiv.1512.03385>) is introduced. This model is employed to regulate QD density during the growth process. However, the model exhibited suboptimal performance in predicting QD density, particularly in relation to QD formation and density labeling. Overall, the concept is not novel, and the data is not convincing to prove the author's concept, making the manuscript unsuitable for publication in Nature Communications.

Here are some comments:

1) In Fig.1, the AFM images correspond to a specific QD density and their RHEED images. Each QD density label fell within a specific density range. What are the criteria to distinguish the label of low, middle, high density? Is it possible to tell the boundary point of each label by the AFM and RHEED images?

2) How many experiments did the authors do for training the database of the model to improve the accuracy? In the Supporting Information, the authors said that there was an insufficient size of the data set. It is hard to understand why the authors did not increase the data set and doubt their model accuracy.

3) The authors claimed that "we demonstrated that ML from previous growth could predict the post-growth density of QDs, by successfully tuning the QD densities in near-real time from $1.5 \times 10^{10} \text{ cm}^{-2}$ down to $3.8 \times 10^8 \text{ cm}^{-2}$ or up to $1.4 \times 10^{11} \text{ cm}^{-2}$ ". However, this statement may lead to misunderstandings that ML can reduce high QDs density to low QDs density in real-time. All experiments started at different initial QDs densities, ranging from zero to low, middle, and high. If the QDs density is initially high, it is not feasible to control it to a lower density using ML in real-time.

4) In fig. 2d, why did the authors choose eight consecutive images for bundling to a 3D matrix. Is it optimized in terms of the data accuracy and the algorithm response time?

5) The authors claimed that RHEED video was analyzing. However, the input was eight consecutive images from the video, not the whole video.

6) In Fig. 3, the authors only mentioned the accuracy of the "QDs model". What about the accuracy of the "density model"? This "density model" is important in determining QDs density. However, the training performance of this model did not appear to be good.

7) Why did the authors choose ResNet 50 as an ML-based model? There was no comparison between the ResNet 50 model and other ML-based models.

8) In Fig. 4, based on the "QDs model", the QDs were just formed, and the In shutter closed almost at the same time, allowing the authors to collect the "low" QDs density of $3.8 \times 10^8 \text{ cm}^{-2}$. If the target QDs density was larger, for example, $1.7 \times 10^9 \text{ cm}^{-2}$, can their software still judge that it is the 'low' label and produce the exact number of the target QDs? I wonder how precise their system is to produce a specific amount of density.

9) In Fig. 4f, despite a significant increase in the probability of the output 'Yes'; the majority of the pie chart represents the output 'No'; Does this mean that the experimental result did not satisfy the 'low' label?

10) In Fig. 5, based on the "QDs model", the authors said that "It is noteworthy that there is still a distance between the blue line and the last "No" label, since we confirm the QD formation by collecting multiple results of the model". Does it mean that their model is not accurate enough?

11) In Fig. 5, the output 'Yes' in the 'QDs model' started at a sequence around 1050, but why did the In shutter close at a sequence around 1400? If the In shutter had closed right after the QDs formation, could 'high' density QDs have been collected? I doubt the feasibility of their model to predict the QD formation.

12) In Fig. S9, the 'density model' prediction results showed an increase in the 'high' label as the target. However, the majority of the results still belonged to the 'low' label, and it did not fit with the target label.

Reviewer #4 (Remarks to the Author):

The primary result of this paper is the use of two convolutional neural network models (ResNet) to analyze RHEED patterns from MBE prepared QD films in real-time. The output of these models is the prediction of QD formation, and the density of synthesized films. These are used to optimize towards two conditions identified by the authors as "high" and "low" density.

In my opinion the results of this approach are not particularly impressive. In the example optimizing towards a low surface density of QDs, the model never reliably predicts quantum dot formation during the optimization based on the contents of Figure 4f, even in the later epochs it is still predicting "no" QD formation in ~80% of cases. Similarly, the density prediction seems to not improve particularly over the course of the optimization either (Figure S8). In the example of the high surface density optimization, where the signal is better the model does apparently accurately predict QD formation, although the density of the QDs synthesized (Figure 5f) are much lower than the target densities (figure 1e&f). Figure S9 seems to suggest that the optimization never reliably finds conditions suitable to produce the high density films. Additionally, I think the article could do with reorganization before publication in any journal, there are details in the main text about the ML model, which are too technical for the main text (but are important and should be included in the SI) and results that are important such as the density prediction model, which are only in the SI.

Regardless, my feeling is that the optimization is never really that successful unfortunately, and that this work is not appropriate for Nature Communications.

I have some additional comments for the authors, which I hope they find helpful.

Abstract

*RHEED is not defined.

*The phrase "specifically designed for training RHEED videos" does not make sense. The ML model is trained 'using' RHEED videos as an input

*"In situ" should not be hyphenated

Line 52

* I don't agree that ML is capable of "establishing complex systems' empirical functions". ML regressors are capable of representing these underlying relationships within a dataset through what is essentially (potentially very high-dimensional) curve-fitting. Obtaining such functions requires an inverse design approach from a well-trained model.

Line 57

* "once the sample characterization results deviate" - deviate from what, please specify

Line 69

* "growers" - would "MBE users" be a better phrasing?

Line 80-81

* for better readability move "automatically and intelligently" before the word achieving.

Line 100-2

* the phrasing here is confusing. It needs to be specified that the authors used AFM to determine that the surface was flat and the label "zero" refers to the density of QDs on the surface.

Line 109

* Remove the word "exceeding", as this definition includes the "high" condition

Figure 1c-f, 4e,S7

* The labels for each condition are not given on the figure or in the caption.

* please provide a scale bar for the heights in the AFM images. Without this the images do not support the claims in the text about flatness. Ideally the Z-scale should be the same for all images that are being compared, which currently it is not.

Figure 1g-j

* The streaks in the RHEED are not very visible when the article is printed out on paper, is it possible to alter the contrast of the images without comprising the data?

Figure 2a

* Personally, I think the representation of the ML model is not great. I think that it would be easier for many readers if the authors clearly represented the inputs and the outputs of the model, rather than the images of the stacked grids and the parts of the MBE with no explanation of what they are supposed to represent. Something like the top part of Figure 6 of this paper, for me is much clearer and useful to the reader. <https://onlinelibrary.wiley.com/doi/full/10.1002/cjoc.202000393> (this is not a request to cite this paper, it is just the first example that I found of a suitable diagram).

Figure 2c

*this figure is redundant, the cropped region could easily be indicated in part b

Line 155

* The Authors should introduce that 3D ResNet 50 Model is the model that they are using at the start of this paragraph, and briefly describe what it does/why it was used, rather than jumping straight into details about which residual blocks they used.

Lines 155-182 and Figure 3

* Following on from this point. The details given in these 2 paragraphs are important, but I feel they are inappropriate in the main text, and would be better placed in the SI for interested readers. I think the level of detail is too high for most readers who will be interested in the implications of this approach on producing materials, rather than the specifics of how the authors set up their CNN. I think that it would be better if this section was replaced with more general explanation of how the machine learning was used. It would also be beneficial to explicitly state which optimization technique you are using here and in the Methods section.

***Why is there no more discussion of the surface density prediction model in the main text after this point? Why is it all hidden in the SI?

Line 224.

* The phrase "significant increase" needs to be quantified. It appears from the pie charts to go from about 15 to about 20 %, which is not so great, if the model is supposed to be accurately predicting QD formation.

Figure 4

* It's hard to resolve the RHEED images when printed out on paper again. If possible it would be better to show a RHEED scattering pattern close to the threshold indicated by the blue line. From the data shown you could only conclude that QD formation occurs somewhere between epoch ~1430 and ~1600

* Finally 4f does not really display very well the number of "yes" and "no" conditions, it seems like "no" happens constantly throughout all epochs because the points overlap each other. I think this format is very poor in almost all instances that it is used, adding a running average across all epochs would demonstrate the changes more clearly to the reader, if there is a major change.

It is also unclear which portions of the graph the two inset pie charts apply to. Is the left one all epochs below the blue line and the right all after?, this needs to be made clearer. This is true for the other instances of pie charts in the main text and SI also.

Line 233-4

* The sentence defining what "yes" or "no" means in relation to the ML model should appear earlier in the paragraph. Preferably on line 223, before the phrase "Before the QD formation..."

Figure 4f

* The AFM image does not look particularly dense compared to the examples of "high density" in figure 1

Figure 4g

* It is not clear to me why the model starts predicting QD formation before the blue line

Line 304

* STAIB is not defined

Line 315-42

* An explanation of the workings of the basic residual block of ResNet works is not appropriate here.

* Important information on the machine learning model is missing from the methods section. There should be information on the software used, how the model was constructed, and which hardware was used. These are all missing.

Dear Reviewers,

We thank the Reviewers for the prompt reply and considering our interdisciplinary research on quantum dots. We deeply appreciate for the critical and constructive review of the manuscript. Reviewers 1 and 2 affirmed the importance of the results in the manuscript for a broad community, and suggested publication as is and publication after minor clarifications, respectively, which we are grateful for the remark. Reviewers 3 and 4 also provide very helpful comments. The suggestions have been fully addressed by incorporating additional required information, thus substantially improved the manuscript. All required changes have been addressed and highlighted in **RED** in the revised manuscript.

Our point-by-point response are included in the following for the convenience of the reviewers:

Reviewer 1:

Q1: The authors can comment how their methodology can be adapted to other QD systems (other than the SK QDs), for example, the droplet QDs, the local droplet etching and manhole infilling QDs.

We have fully addressed the comment.

The droplet QDs are indeed another important QD system. The formation of droplet QDs, the local droplet etching and manhole infilling QDs have been studied by Prof. Nobuyuki Koguchi (Katsuyuki Watanabe et al; Fabrication of GaAs Quantum Dots by Modified Droplet Epitaxy. Jpn. J. Appl. Phys. 2000; 39 L79); Prof. Zh. M. Wang (Zh. M. Wang, B. L. Liang, K. A. Sablon, G. J. Salamo; Nanoholes fabricated by self-assembled gallium nanodril on GaAs(100). Appl. Phys. Lett. 2007; 90 (11): 113120); Dr. Christian Heyn (Ch. Heyn, A. Stemmann, T. Köppen, Ch. Strelow, T. Kipp, M. Grave, S. Mendach, W. Hansen; Highly uniform and strain-free GaAs quantum dots fabricated by filling of self-assembled nanoholes. Appl. Phys. Lett. 2009; 94 (18): 183113). The RHEED

tracking of these structures has also been demonstrated with distinct features (Á. Nemcsics, C. Heyn, A. Stemmann, A. Schramm, H. Welsch, W. Hansen, The RHEED tracking of the droplet epitaxial grown quantum dot and ring structures, Mater. Sci. Eng. B., 2009, 165, 118-121.).

Our methodology can also be adapted to capture the RHEED features of these QDs formation, train the ML models, and provide feedback to tune the growth parameters in real-time. So, the processes can be controlled to achieve expected structures.

We included the following comments, “The methodology can also be adapted to other QD systems, such as the droplet QDs, the local droplet etching, and manhole infilling QDs with distinct RHEED features.” on **Page 6, Line 105** to **Page 6, Line 107** of the **manuscript**.

The mentioned references are also cited in the manuscript.

Reviewer 2:

Q1: More detail should be discussed about the Resnet algorithms used. As written the unfamiliar reader might interpret the authors as originating this structure. Instead, the authors merely applied this structure to show how it can be used to learn and predict QD density. The authors should clarify any specific changes relative to the standard Resnet approach and make clear any differences they innovated or if they merely used the pre-existing image processing approach.

We apologize for any ambiguity that may have led to a misunderstanding regarding the ResNet algorithm used in our study. Our model shares a similar structure with the standard ResNet model but incorporates several key modifications to adapt it to the specific needs of learning and predicting quantum dot (QD) density:

Firstly, we employ 3D convolution instead of the 2D convolution used in the standard ResNet model. Secondly, we’ve adjusted the structure of identity shortcuts to reduce information loss during downsampling. Thirdly, we’ve reduced the frequency at which the

model doubles the number of channels after passing through the residual structure. Finally, we also provided a detailed description of the parameters of each convolution used in the model.

To make it clear, we included detailed discussion about the ResNet 50 algorithms used and clarify any specific changes relative to the standard Resnet approach as follows: “When processing video frame data, 3D convolution proves more effective in extracting spatial and temporal dimension information from videos than the 2D convolution used in the standard ResNet model. The difference between our network and the original ResNet lies in the dimensionality of the convolutional kernel and batch normalization operations. Our 3D ResNet 50 model employs 3D convolutions and 3D batch normalization. This spatiotemporal-feature-learning method enables the 3D ResNet 50 model to better capture the dynamic information in videos compared to 2D CNN, resulting in an enhanced ability to identify and differentiate the categories of videos. Furthermore, we have adjusted the structure of identity shortcuts to reduce information loss during downsampling and reduced the frequency at which the model doubles the number of channels after passing through the residual structure.” on **Page 11, Line 186 to Page 11, Line 195** of the manuscript.

The model’s residual block structure, specific parameter information for each convolution, and overall framework information can be found in S5 titled “The detailed structure of the ResNet 50 model” on **Page 8, Line 100 to Page 11, Line 144** and Fig. S7 of the supplementary information.

Q2: Some discussion of data size and throughput (speed) management should be included. Having done some of these calculations, I am astonished that a 8x300x300 pixel image frame video can be processed real time without data reductions like found in PCA for example. I am even more astonished this can be done within the notoriously slow LabVIEW environment. This discussion should include the computational difficulty and speed of the training process and separately, the real time inference. These limitations are the key to practical implementation. Finally, it is well known that each MBE system has physical limitations in the wobble of substrates. Each tool is different. The authors should address

the universal applicability of the developed model. Can their model be applied to other systems with different wobble characteristics or will a new training cycle be required for every tool. In general, I find this method more useful for production MBE than research owing to the repetitive nature of production versus the day-to-day variance of research activities. nevertheless, it is an impressive display of ML power.

We thank the reviewer for appreciating our work. We agree that the data size, speed management, and the universal applicability of the model should be discussed. Our responses are as follows:

(A) Some discussion of data size and throughput (speed) management should be included.

The discussion of data size and speed management are essential for our manuscript. We have implemented data preprocessing before the model training process. Before initiating the training process, we converted the original video data into individual images with a frame resolution of 450×450 pixels. We compared the model's accuracy when utilizing different numbers of images, as illustrated in Fig. R1. Our findings indicate that when selecting more than 8 images, the model accuracy shows little improvement. So, we applied image augmentation techniques to convert these image sequences to $8 \times 300 \times 300$ pixels into NumPy arrays, with each sample approximately 704 KB in size.

Fig. R1: (a) The QDs model and (b) the density model train and valid accuracy with different numbers of images.

To make it clear, we included the following sentence, “We then compared the impact of choosing different numbers of images on model training accuracy and speed, ultimately determined that selecting 8 images was the appropriate choice.” on **Page 9, Line 169** to **Page 10, Line 171** of the manuscript.

“Finally, we conducted preprocessing for all data before model training to reduce the time. We have also applied data augmentation techniques to the training data, including adjusting image curves, cropping images, and scaling images, which effectively increase data diversity and improve the model’s generalization ability. This process resulted in the acquisition of approximately 360,000 NumPy arrays with each sample approximately 704 KB in size.” on **Page 10, Line 172** to **Page 10, Line 176** of the manuscript.

As shown in Fig. R2, for the model training and validation process without data preprocessing, we saw a considerable reduction in processing speed. More precisely, the training speed decreased from 89 to 14 samples per second, and the validation speed dropped from 136 to 15 samples per second. This decline in processing speed presented significant challenges in effectively training the model and significantly extended the time

required for the training process.

Fig. R2: Model training and validation speed before and after preprocessing with different numbers of images.

We have also provided a discussion of the number of images selected in S3 titled “The number of images selected” on **Page 6, Line 74** to **Page 6, Line 79** and Fig. S5 of **the supplementary information**, and S4 titled “Speed management” on **Page 7, Line 85** to **Page 8, Line 94** and Fig. S6 of **the supplementary information**.

(B) **Having done some of these calculations, I am astonished that a 8x300x300 pixel image frame video can be processed real time without data reductions like found in PCA for example.**

PCA is a linear method in feature engineering that is applied to reduce the dimensionality of data while preserving the information of the original data to the greatest extent possible. ResNet 50 has the capability to automatically learn methods for extracting valuable features from raw data without the need for manual feature engineering or dimensionality reduction, which sets it apart from traditional PCA. For a 8x300x300 pixel image frame video, both QDs model and the density model can each produce results at an approximate rate of 1 sample per second, effectively meeting the application’s requirements for process in real time.

To make it clear, we included the following sentence, “The ResNet 50 model has

emerged as our preferred choice for several reasons, including its capability to address gradient vanishing issues and automatically learn methods for extracting features from raw data without manual feature engineering or dimensionality reduction, rapid convergence, and suitability for deep training.” on **Page 11, Line 183** to **Page 11, Line 186** of the **manuscript**.

(C) I am even more astonished this can be done within the notoriously slow LabVIEW environment. This discussion should include the computational difficulty and speed of the training process and separately, the real time inference. These limitations are the key to practical implementation.

We discovered that with preprocessing in place, the model’s training and validation speeds not only doubled compared to the instances where no data preprocessing was applied but also showed stable speed in response to changes in the number of images, as shown in Fig. R2. Moreover, before deploying the model in the LabVIEW software, we converted it into a universal Open Neural Network Exchange (ONNX) format. This conversion enabled us to use a LabVIEW plugin for implementation, integrating the TensorRT accelerator into the model processing pipeline. Consequently, both the QDs model and the density model can each output nearly 1 result per second, which is enough for real time inference.

To make it clear, we included the following sentence, “Moreover, we converted the model into the open standard Open Neural Network Exchange (ONNX) format. When deployed in a LabVIEW environment equipped with TensorRT accelerators, the QDs and density models can each produce results at an approximate rate of 1 sample per second, effectively meeting the application’s requirements.” on **Page 12, Line 225** to **Page 12, Line 228** of the **manuscript**, and S4 titled “Speed management” on **Page 7, Line 85** to **Page 8, Line 94** and Fig. S6 of the **supplementary information**.

(D) Finally, it is well known that each MBE system has physical limitations in the

wobble of substrates. Each tool is different. The authors should address the universal applicability of the developed model. Can their model be applied to other systems with different wobble characteristics or will a new training cycle be required for every tool.

We thank the reviewer for raising this important question.

The wobble characteristic is indeed a physical limitation for each MBE system, which is important to the practical applicability of our method, and we also considered in our research. The wobble of substrates will be observed in the RHEED videos took during the growth. Using image augmentation techniques based on existing data, we conducted simulations and generated data with different wobble patterns in both horizontal and vertical directions.

The mechanical vibration is a specific manifestation of substrate wobbling, which we can simulate during data preprocessing. In addition to the image augmentation methods based on color curves and pixel scaling, we also perform image cropping based on the original data. The offset of the image cropping selection from the edge is crucial in simulating mechanical vibration. To achieve this, we randomly generate a sine function with a period matching the substrate rotation period. The amplitude of the sine function is generated using Python's random numbers. Then, we randomly set a starting point within a single cycle and collect 24 points within the length of one cycle based on this starting point. Afterward, 8 consecutive points are selected from these 24 points as the cropping offset in the horizontal or vertical direction. Each cropped image is then resized to 300×300 pixels. This process creates a new dataset of 8 images, each at 300×300 pixels. Consequently, this dataset exhibits the characteristic of substrate wobbling, which was subsequently employed for fine-tuning model parameters without a new training cycle, resulting in an accuracy of 98.8% of QDs model and 98.4% of density model after 5 epochs, as shown in Fig R3. Therefore, our model exhibits universal applicability in MBE systems with different wobble characteristics.

Fig. R3: Training results of fine-tuning the (a) QDs model and (b) density model on wobble characteristic data.

To make it clear, we include the sentence “Additionally, the model is universally applicable to other systems with different wobble characteristics without a new training cycle.” on **Page 13, Line 230** to **Page 13, Line 231** of the manuscript.

“Our model also exhibits potential for application in MBE systems with different wobble characteristics. The mechanical vibration is a specific manifestation of substrate wobbling, which we can simulate during data preprocessing. Using image augmentation techniques based on color curves and pixel scaling using existing data, we conducted simulations and generated data with different wobble patterns in both horizontal and vertical directions. In addition, we also perform image cropping. The offset of the image cropping selection from the edge is crucial in simulating mechanical vibration. To achieve this, we randomly generate a sine function with a period matching the substrate rotation. The amplitude of the sine function is generated using Python’s random numbers. Then, we randomly set a starting point within a single cycle and collect 24 points within the length of one cycle based on this starting point. Afterward, 8 consecutive points are selected from

these points as the cropping offset in the horizontal or vertical direction. Each cropped image is then resized to 300×300 pixels. This process results in the creation of a new dataset consisting of 8 images. Consequently, this dataset exhibits the characteristic of substrate wobbling, which was subsequently employed for fine-tuning model parameters without a new training cycle, resulting in an accuracy of 98.8% of QDs model and 98.4% of density model after 5 epochs, as shown in Figure S13.” in S11 titled “The QDs and density models with different wobble characteristics” on **Page 16, Line 216 to Page 16, Line 231** and Fig. S13 of **the supplementary information**.

Reviewer 3:

Q1: Numerous studies have explored the regulation of InAs quantum dot (QD) growth density through the MBE experiments, as evidenced by works such as <https://doi.org/10.1063/1.1327613>. Additionally, scholars have investigated feedback control in MBE experiments ([https://doi.org/10.1016/S0952-1976\(98\)00024-4](https://doi.org/10.1016/S0952-1976(98)00024-4)) and alternative optimization models for handling reflection high-energy electron diffraction (RHEED) video in film growth processes (<https://doi.org/10.1186/s40580-023-00359-5>). In this study, a novel approach utilizing the 3D ResNet 50 model (originating from the ResNet model:<https://doi.org/10.48550/arXiv.1512.03385>) is introduced. This model is employed to regulate QD density during the growth process. However, the model exhibited suboptimal performance in predicting QD density, particularly in relation to QD formation and density labeling.

Thank the Reviewer for the suggestions. We have fully addressed the comments as follows:

(A) Numerous studies have explored the regulation of InAs quantum dot (QD) growth density through the MBE experiments, as evidenced by works such as <https://doi.org/10.1063/1.1327613>.

In this paper (<https://doi.org/10.1063/1.1327613>), the authors controlled the growth density of QDs by optimizing the mask pattern and growth conditions. Compared to automated optimization through control systems (the advance of proposed approach in our manuscript), this existing method requires time-consuming trial-and-error conducted by skilled growers, as mentioned in the introduction of our manuscript: “Time-consuming trial-and-error testing is inevitably required to establish optimal process parameters for the intended material specification.” on **Page 4, Line 63** to **Page 4, Line 65** of the manuscript.

(B) Additionally, scholars have investigated feedback control in MBE experiments ([https://doi.org/10.1016/S0952-1976\(98\)00024-4](https://doi.org/10.1016/S0952-1976(98)00024-4)) and alternative optimization models for handling reflection high-energy electron diffraction (RHEED) video in film growth processes (<https://doi.org/10.1186/s40580-023-00359-5>).

In this paper ([https://doi.org/10.1016/S0952-1976\(98\)00024-4](https://doi.org/10.1016/S0952-1976(98)00024-4)), an ellipsometer was utilized for in-situ monitoring of material composition, and the feedback was employed to regulate the cell temperatures. This existing method adjusts its growth parameters of thick layers after material composition deviation already happens. Moreover, this can be detrimental for complicated film structures with extremely short growth durations as mentioned by the authors of this paper. In contrast, our method focuses on predicting whether the growth outcomes align with expectations before it happens and actively involves in regulating material parameters at early stage of the growth process. This early intervention can effectively mitigate the occurrence of material quality deviation.

To make it clear, we have reorganized this paragraph and included the following sentence, “after an amount of film being deposited” on **Page 4, Line 76** to **Page 5, Line 77** of the manuscript and “To enable early intervention in the growth process before the QDs formation, we developed a control logic for the LabVIEW program, as shown in Figure 3d.” on **Page 13, Line 233** to **Page 13, Line 234** of the manuscript.

In the paper (<https://doi.org/10.1186/s40580-023-00359-5>), ML-assisted analysis of

RHEED videos aids in interpreting the RHEED data of oxide thin films after the material growth was completed. In our research, we implement real-time analysis of RHEED videos during the material growth process, enabling us to proactively adjust material growth parameters based on the ongoing results at each moment.

The paper (Kim H J, Chong M, Rhee T G, et al.; Machine-learning-assisted analysis of transition metal dichalcogenide thin-film growth. *Nano Convergence*. 2023; 10 (1): 10) is also cited on **Page 4, Line 71** of the manuscript.

(C) In this study, a novel approach utilizing the 3D ResNet 50 model (originating from the ResNet model:<https://doi.org/10.48550/arXiv.1512.03385>) is introduced.

The standard ResNet model was introduced in this article (<https://doi.org/10.48550/arXiv.1512.03385>), our model shares a similar structure with the standard ResNet model but differs in its detailed design. Firstly, we employ 3D convolution instead of the 2D convolution used in the standard ResNet model. Secondly, we've adjusted the structure of identity shortcuts to reduce information loss during downsampling. Thirdly, we've reduced the frequency at which the model doubles the number of channels after passing through the residual structure. Finally, we also provided a detailed description of the parameters of each convolution used in the model.

To make it clear, we included the following sentence, "When processing video frame data, 3D convolution proves more effective in extracting spatial and temporal dimension information from videos than the 2D convolution used in the standard ResNet model. The difference between our network and the original ResNet lies in the dimensionality of the convolutional kernel and batch normalization operations. Our 3D ResNet 50 model employs 3D convolutions and 3D batch normalization. This spatiotemporal-feature-learning method enables the 3D ResNet 50 model to better capture the dynamic information in videos compared to 2D CNN, resulting in an enhanced ability to identify and differentiate the categories of videos. Furthermore, we have adjusted the structure of identity shortcuts to reduce information loss during downsampling and reduced the frequency at which the

model doubles the number of channels after passing through the residual structure.” on **Page 11, Line 186 to Page 11, Line 195 of the manuscript.**

The mentioned article is also cited in the manuscript.

(D) This model is employed to regulate QD density during the growth process. However, the model exhibited suboptimal performance in predicting QD density, particularly in relation to QD formation and density labeling.

The density model training results are displayed in Figure 3c in the manuscript, although there are still some fluctuations, the average validation accuracy of the last 10 epochs is 95.1%.

We appreciate the Reviewer raised questions about the pie charts we included in the original manuscript; we have updated them according to reviewers' suggestions to better show the performance of our method.

To effectively depict data trends, we have adjusted pie charts in the manuscript. The statistical analysis conducted on the initial 200 sequences is intended to verify whether the initial growth conditions are capable of producing QDs with the predetermined target density. The subsequent analysis, spanning from the 200th sequence to 40 sequences before QD formation, serves as a comparative examination against the initial 200 sequences. This stage helps ascertain whether the material growth conditions have been progressively adjusted to better align with the desired growth conditions. The 40 sequences before QD formation are analyzed to demonstrate the QD model's potential to detect the initiation of QD formation. Finally, an analysis of the 40 sequences after QD formation is conducted to affirm the successful formation of QDs and to portray the ultimate outcome and density change trend of the QDs.

To make it clear, we have updated all pie charts in the manuscript and included the following sentence “To effectively depict data trends, we have incorporated four pie charts throughout to illustrate the evolution of the model's output results, covering a statistical range spanning the initial 200 sequences, from the 200th sequence to 40 sequences before

QD formation, the final 40 sequences before QD formation, and 40 sequences after QD formation.” on **Page 15, Line 267** to **Page 16, Line 270** of the manuscript.

Using the pie chart method described above, the modified original Fig. S8 is displayed as Fig. R4. When examining the proportion of “low” labels in the four pie charts, as the sequence advances, the likelihood of the density model outputting “low” gradually rises. This indicates that growth conditions are more favorable for the growth of low-density QDs after the gradual adjustment.

Fig. R4. The prediction results of the “density model” in the experiment targeting the “low” label, blue line: the QD formation time; yellow line: the In shutter closing time.

To make it clear, we have relocated original Fig. S8 of the supplementary information to Fig. 4g of the manuscript and included the following sentence, “Figure 4g shows the “density model” results of the experiment with the “low” label as the target. In the first pie chart, the probability of output “low” is significantly lower than that of combined “middle” and “high”. However, with the real-time feedback control of growth conditions, it is evident in the second pie chart that the probability of output “low” approaches 50%. The growth conditions have been gradually adjusted to better suit the growth of low-density QDs. It becomes apparent from the third and fourth pie charts that the probability of the model outputting “low” surpasses 55% and persists until the end of the growth.” on **Page 16, Line 280** to **Page 16, Line 286** of the manuscript. Additionally, we have incorporated this citation into the manuscript.

In high-density growth experiments, the shutter remains open after the formation of QDs for a period. Therefore, we added two additional pie charts, one before and one after

the shutter is closed, to further illustrate the results of the density model.

To make it clear, we included the following sentence, “In high-density growth experiments, the shutter remains open after the formation of QDs for a period. Therefore, we incorporated two additional pie charts: one for the 40 sequences before the shutter closure and another for the 40 sequences after the shutter closure.” on **Page 18, Line 307** to **Page 18, Line 309** of the manuscript.

Utilizing the pie chart data method described earlier, the adjusted results of original Fig. S9 are also presented in Fig. R5. We can clearly observe that before the formation of QDs, the model outputs only a small proportion of “high” labels. After the formation of QDs, the probability of “high” label output rapidly increases, and the probability of “high” label output is much higher than other labels before and after the shutter is closed. The probability change of “high” label output from the density model fully demonstrates the process of growth environment adjustment from not suitable for high-density QD growth to gradually favorable for high-density QD growth.

Fig. R5: The prediction results of the “density model” in the experiment targeting the “high” label, blue line: the QD formation time; yellow line: the In shutter closing time.

To make it clear, we have relocated original Fig. S9 of the supplementary information to Fig. 5h of the manuscript and included the following sentence, “The output results of the density model are also presented in Figure 5h. Before QDs formation, the density model rarely outputs “high”. However, as the growth process approaches the stage of QD formation, the probability of “high” label output slowly increases to 15%, as demonstrated in the first to third pie charts. Following the formation of QDs, the probability of the density model outputting “high” rapidly increases to 30%, as depicted in the fourth pie chart. Due

to the unique nature of QD systems, closing the shutter at this point would restrict the formation of high-density QDs. (Placidi E, Arciprete F, Fanfoni M, et al. InAs/GaAs (001) epitaxy: kinetic effects in the two-dimensional to three-dimensional transition. Journal of Physics: Condensed Matter, 2007, 19(22): 225006; Kobayashi N P, Ramachandran T R, Chen P, et al. In situ, atomic force microscope studies of the evolution of InAs three-dimensional islands on GaAs (001). Applied physics letters, 1996, 68(23): 3299-3301) As the growth process continues and approaches the moment the shutter is closed, it becomes evident that the probability of the model outputting “high” rapidly increases to over 70%, illustrated in the fifth and sixth pie charts.” on **Page 18, Line 317** to **Page 19, Line 325** of **the manuscript**. Additionally, we have included citations in the manuscript.

Q2: In Fig.1, the AFM images correspond to a specific QD density and their RHEED images. Each QD density label fell within a specific density range. What are the criteria to distinguish the label of low, middle, high density? Is it possible to tell the boundary point of each label by the AFM and RHEED images?

We have fully addressed the comment.

We defined the boundaries of different labels based on the results of AFM and the features of RHEED patterns. After the formation of QDs, the RHEED patterns corresponding to the “low” label exhibit a coexistence of streak and spot features, with a QD density of less than $1 \times 10^{10} \text{ cm}^{-2}$.

For the “middle” label, the RHEED patterns display spot features with a QD density ranging from greater than $1 \times 10^{10} \text{ cm}^{-2}$ to less than $4 \times 10^{10} \text{ cm}^{-2}$.

In the case of the “high” label, the RHEED patterns also show spot features, but in comparison to the “middle” label, the spots are brighter and more rounded. Moreover, the corresponding QD density exceeds $4 \times 10^{10} \text{ cm}^{-2}$.

To make it clear, we included the following sentence, “Upon conducting a more thorough analysis of the RHEED patterns, we identified the patterns transition from having both streaks and spots to only spots, corresponding to a density of $1 \times 10^{10} \text{ cm}^{-2}$ in AFM images.

So, we assigned the “low” label to densities below $1 \times 10^{10} \text{ cm}^{-2}$, the “middle” label to densities ranging from $1 \times 10^{10} \text{ cm}^{-2}$ to $4 \times 10^{10} \text{ cm}^{-2}$. As the density exceeds $4 \times 10^{10} \text{ cm}^{-2}$, the spot features become more rounded and show higher brightness, we labeled densities exceeding $4 \times 10^{10} \text{ cm}^{-2}$ as “high.” on **Page 7, Line 127** to **Page 7, Line 132** of the **manuscript**.

Q3: How many experiments did the authors do for training the database of the model to improve the accuracy? In the Supporting Information, the authors said that there was an insufficient size of the data set. It is hard to understand why the authors did not increase the data set and doubt their model accuracy.

We have fully addressed the comments as follows:

(A) How many experiments did the authors do for training the database of the model to improve the accuracy?

To improve the model’s accuracy, we did the following experiments for training the database of the model.

Firstly, as depicted in the original Fig. 1b of the manuscript, we prepared 30 samples with different QD densities. We repeated the QD growth on average 4 times using the same parameters for each sample, resulting in a total of 120 growth experiments with video datasets.

Secondly, during the data preprocessing phase, the sequences within each video are sequentially concatenated to create training samples. Subsequently, these training samples undergo random image enhancements, which include modifying the image curves to adjust color saturation and brightness, random image cropping, and image scaling. These enhancements are implemented to fully leverage the existing video data.

To make it clear, we included the following sentence, “Each QD growth was repeated on average 4 times, leading to the generation of 120 RHEED videos.” on **Page 6, Line 118** to **Page 6, Line 119** of the **manuscript**.

“We have also applied data augmentation techniques to the training data, including adjusting image curves, cropping images, and scaling images, which effectively increase data diversity and improve the model’s generalization ability.” on **Page 10, Line 173** to **Page 10, Line 176** of the manuscript.

Moreover, we trained different models 16 times to ensure and enhance model accuracy. This training regimen consisted of 14 model training sessions using different preprocessed data to explore the optimal quantity of preprocessed images, 1 model training session based on the ResNet 50 encoder-decoder structure to ensure effective feature extraction from the preprocessed data, and 1 exploration to determine whether a single model can concurrently address the issues of discriminative QD formation and QD density. The details are as follows:

We conducted a comparison of the model’s accuracy when using different numbers of images, as shown in Fig. R6. We selected 1, 2, 4, 6, 8, 10, and 12 images to construct various training samples, resulting in the creation of multiple datasets. Therefore, we conducted a total of 14 model training sessions for both the QDs model and the density model. It is evident that as the number of selected images increases, the validation accuracy of both the QDs model and density model gradually improves. However, achieving further improvement in model accuracy becomes challenging when selecting more than 8 images.

Fig. R6: (a) The QDs model and (b) the density model train and valid accuracy with different numbers of images.

To validate that the model possesses adequate complexity for data feature extraction, we trained an encoder-decoder architecture network based on ResNet 50. After 100 epochs, the model achieved an impressive low loss value of 0.0041746, as shown in Fig. R7.

Fig. R7: Training results of encoder decoder architecture model based on 3D ResNet 50.

To enhance efficiency, we then attempted to train a single model to predict five classification results, including one label determining the QD formation and four density labels. The accuracy plateaued at around 96.3%, as shown in Fig. R8. Analyzing data with misclassifications posed further challenges in pinpointing the sources of these errors. It is difficult to find a unified pattern for these misclassified data in the dataset to optimize it. Finally, we implemented separate density model and QDs model.

Fig. R8: The training and validation data for the 3D ResNet 50 model consist of five categories: one to determine whether QDs have formed and four for density labels.

To make it clear, these can be found in S3 titled “The number of images selected” on **Page 6, Line 74** to **Page 6, Line 79** and Fig. S5 of **the supplementary information**, and S8 titled “Model development and training results” on **Page 13, Line 189** to **Page 14, Line 202** and Fig. S9-10 of **the supplementary information**.

(B) **In the Supporting Information, the authors said that there was an insufficient size of the data set. It is hard to understand why the authors did not increase the data set and doubt their model accuracy.**

We iteratively optimized the data labels, ultimately achieving validation accuracies of 95.1% for the density model and 94.4% for the QDs model. In our original supplementary materials, we emphasized the importance of expanding the dataset and exploring precise label partitioning methods simultaneously. As of now, achieving a more effective label partitioning method by further subdividing the labels in our research remains a challenge. Consequently, under the current label partitioning methods, even with dataset expansion, the model’s training results will continue to be influenced by the difficulty of accurately partitioning labels.

We further compared the output results of the density model using 10% data volume with the output results using 100% data volume using the same label partitioning method, the model accuracy did not increase with the data volume. Specifically, during the training process of the density model, we used 10% of the data to test, and the validation accuracy when selecting 8 images as samples reached 99.1%. However, when using the complete dataset for training, the accuracy of the density model can only reach a maximum of 95.1%. The results of the QDs model are also similar. When trained with a 10% dataset, the model’s validation accuracy can reach over 97%; but when trained with the complete dataset, the accuracy maintains 94.4%. This result indicates that solely increasing the amount of data without optimizing the label partitioning method cannot consistently improve model performance.

To make it clear, we rewrite and included the following sentence, “Moreover, as the

breakthroughs in data label partitioning methods and dataset expansion in the future, we anticipate achieving finer-grained density regulation during the material growth process.” from the supplementary information and placed it on **Page 20, Line 345** to **Page 20, Line 346** of **the manuscript**. We also included S7 titled “Comparison of training and validation results of different models” on **Page 12, Line 176** to **Page 13, Line 184** and Fig. S8 of **the supplementary information**.

Q4: The authors claimed that “we demonstrated that ML from previous growth could predict the post-growth density of QDs, by successfully tuning the QD densities in near-real time from $1.5 \times 10^{10} \text{ cm}^{-2}$ down to $3.8 \times 10^8 \text{ cm}^{-2}$ or up to $1.4 \times 10^{11} \text{ cm}^{-2}$ ”. However, this statement may lead to misunderstandings that ML can reduce high QDs density to low QDs density in real-time. All experiments started at different initial QDs densities, ranging from zero to low, middle, and high. If the QDs density is initially high, it is not feasible to control it to a lower density using ML in real-time.

Stimulated by the Reviewer’ comment, we have performed new experiment to employ machine learning for real-time control to reduce QDs density when the initial growth conditions are favorable for high-density QDs.

As shown in Fig. R9, we configured the initial growth conditions in the additional experiment to favor high-density QDs while setting the “low” label as the target. As illustrated in Figure R9a, the substrate temperature underwent a total change of 33 °C during the growth, indicating that the initial substrate temperature was not suitable for achieving the “low” label. Furthermore, the convergence of the blue and yellow lines at the same sequence number suggests that, upon QD formation, the “density model” promptly recognized the current RHEED sequence as consistent with the “low” label and terminated the growth process. In Figure R9b, RHEED images primarily displayed streaks during the initial growth stage, even before QD formation, as shown in Figure R9c. After QD formation, distinctive streaks and spot features promptly emerged, aligning with the characteristics of the “low” label, as depicted in Figure R9d. The density of these QDs was

$9.5 \times 10^8 \text{ cm}^{-2}$, with an average diameter of 54.5 nm and a height of 8.3 nm, as evidenced in the AFM image in Figure R9e. Figures R9f and R9g illustrate the distribution of outputs for the QDs model and the density model across the entire sequence. In Figure R9f, the first and second pie charts consistently display a low probability of outputting “Yes”, remaining at 30%. However, as the sequences approach the QD formation, in the third and fourth pie charts, a distinct trend emerges: the probability of the QDs model outputting “Yes” gradually surpasses 72%. Furthermore, in Figure R9g, the pie charts illustrate the progressive shift in the “low” label. In the first pie chart, the presence of the “low” label in the model’s output is 23%. However, the likelihood of the model outputting the “low” label in the second pie chart gradually increases, surpassing 33%. The density model consistently produces “low” outputs in the two subsequent pie charts near QDs formation, finally reaching 62%. It signifies that, under the current conditions, the material is highly inclined to foster the growth of low-density QDs.

Fig. R9: Controlled growth process of low-density QDs. Experiment with the “low” label as the target. (a) Substrate temperature during growth. The RHEED image (b) captured at 200th frame after growth; (c) before the QD formation; (d) after the QD formation; (e) the 2 $\mu\text{m} \times 2 \mu\text{m}$ AFM image of the sample; the prediction results of (f) the “QDs model” and the “density model”.

To make it clear, we included the following sentence, “Furthermore, we also set the initial conditions favorable for the growth of high-density QDs and conducted experiments with the “low” label as the target.” on **Page 20, Line 335** to **Page 20, Line 336** of the **manuscript**.

The discussion can be found in S14 titled “Controlled growth of low-density QDs with initial high-density growth conditions” on **Page 21, Line 281** to **Page 21, Line 302** and Fig. S16 of the **supplementary information**.

Q5: In Fig. 2d, why did the authors choose eight consecutive images for bundling to a 3D matrix. Is it optimized in terms of the data accuracy and the algorithm response time?

We have fully addressed the comment.

We have chosen a number of data to evaluate the training performance of the model with varying numbers of images as shown in Fig. R6. It is evident that as the number of selected images increases, the validation accuracy of both the density model and QDs model gradually improves. However, further improvement of model accuracy becomes not obvious when selecting more than 8 images.

Additionally, we optimized the training and validation process speeds with different numbers of images in density model, as illustrated in Fig. R10. It is evident that incorporating data preprocessing during the training process significantly reduces the data processing speed of the model. However, the data processing speed of the model trained after preprocessing remains relatively constant. Consequently, during model deployment, we also implemented direct data preprocessing through LabVIEW, allowing both models

to output results nearly every second throughout the deployment process.

Fig. R10: Model training and validation speed before and after preprocessing with different numbers of images.

To make it clear, we included the following sentence, “We then compared the impact of choosing different numbers of images on model training accuracy and speed, ultimately determined that selecting 8 images was the appropriate choice.” on **Page 9, Line 169** to **Page 10, Line 171** of the manuscript. We have also provided a discussion of the number of images selected in S3 titled “The number of images selected” and Fig. S5, on **Page 6, Line 74** to **Page 6, Line 79** and Fig. S5 of the supplementary information, and discussed speed management in S4 titled “Speed management” and Fig. S6, on **Page 7, Line 85** to **Page 8, Line 95** and Fig. S6 of the supplementary information.

Q6: The authors claimed that RHEED video was analyzing. However, the input was eight consecutive images from the video, not the whole video.

We have fully addressed the comment.

We chose to combine eight consecutive images into a 3D matrix for several reasons. Due to the symmetry of the GaAs crystal structure, the RHEED images we obtain also show symmetry. When the processed data exceeds the minimum periodicity of lattice symmetry, the model processes an excessive amount of redundant information, which hinders the rapid improvement of model accuracy.

Through systematic inter-frame conversion and information fusion, we can reconstruct the dynamic content of a video from temporal images. As a result, temporal image processing can be regarded as a means of breaking down and abstracting video processing, offering a more adaptable and efficient approach for the analysis and manipulation of information in dynamic scenarios (Wang Y, Xu Z, Wang X, et al. End-to-end video instance segmentation with transformers. Proceedings of the IEEE/CVF conference on computer vision and pattern recognition. 2021: 8741-8750).

To make it clear, the discuss can be found in “RHEED videos taken during growth were first deconstructed into multiple temporal images and utilized as input for our model, which is a technique for breaking down video processing, offering a more versatile and efficient approach to analyze and process information in dynamic scenes.” on **Page 9, Line 150** to **Page 9, Line 152 of the manuscript**. Additionally, we have included this citation in the manuscript.

Q7: In Fig. 3, the authors only mentioned the accuracy of the “QDs model” . What about the accuracy of the “density model” ? This “density model” is important in determining QDs density. However, the training performance of this model did not appear to be good.

We agree with the reviewer on that “density model” is important in determining QDs density. Actually, we mentioned the accuracy of the “QDs model” and “density model” in original Fig. 3. The average validation accuracy of the density model reached 95.1%, as shown in the original Fig. 3i. The data fluctuation is attributed to the occurrence of label overlap during the data label partitioning process. Specifically, the density of QDs gradually increases with deposition, however, there might be a subsequent decrease in QD density due to ripening of the QDs (Tatebayashi J, Nishioka M, Someya T, et al. Area-controlled growth of InAs quantum dots and improvement of density and size distribution. Applied Physics Letters, 2000, 77(21): 3382-3384; Frigeri P, Nasi L, Prezioso M, Seravalli L, Trevisi G, Gombia E, Mosca R, Germini F, Bocchi C, Franchi S. Effects of the quantum dot ripening in high-coverage InAs/GaAs nanostructures. Journal of Applied Physics, 2007,

102(8)).

To make it clear, we included more discussion on **Page 12, Line 208** to **Page 12, Line 220** of **the manuscript**: “Throughout the model training process, the persistent fluctuations and plateaus in the “QDs model” and “density model” accuracy were attributed to the data diversity and the manual label assignment process instead of limitations inherent in the model itself. After 120 training cycles, the “QDs model” maintained a validation accuracy of over 90%, as shown in Figure 3b, ultimately achieving a final average validation accuracy of 94.4%. The fluctuations observed in the “QDs model” validation process predominantly originate from a specific time before the QD formation when the RHEED streaks still dominated most of the collected images. As the QDs are about to form, the spot features closely associated with QD formation are often faint and overshadowed by the streak features. As depicted in Figure 3c, the validation accuracy of the “density model” continues to exhibit fluctuations even after 100 epochs, primarily attributed to label overlap when low-density QD data exhibit an increase in density with deposition. (Tatebayashi J, Nishioka M, Someya T, et al. Area-controlled growth of InAs quantum dots and improvement of density and size distribution. *Applied Physics Letters*, 2000, 77(21): 3382-3384; Frigeri P, Nasi L, Prezioso M, Seravalli L, Trevisi G, Gombia E, Mosca R, Germini F, Bocchi C, Franchi S. Effects of the quantum dot ripening in high-coverage InAs/GaAs nanostructures. *Journal of Applied Physics*, 2007, 102(8)”, “However, it consistently maintained a validation accuracy of over 80%, and the final average validation accuracy reached 95.1%.”

Moreover, we have also discussed the performance of the density model in S8 and S9, which we have improved the pie chart and relocated them to Fig. 4g and 5h, respectively. Figure 4g shows the “density model” results of the experiment with the “low” label set as the target. As growth progresses, it becomes apparent that the probability of the model outputting “low” surpasses 50% and can persist until the end of the growth process. So, the growth conditions gradually adjust and tend towards low-density quantum dot growth. In Figure 5h, as the growth process continues and approaches the moment when the shutter is closed, it becomes evident that the probability of the model outputting “high” rapidly

increases to over 70%, as illustrated in the fifth and sixth pie charts.

To make it clear, the discuss can be found in “Figure 4g shows the “density model” results of the experiment with the “low” label as the target. In the first pie chart, the probability of output “low” is significantly lower than that of combined “middle” and “high”. However, with the real-time feedback control of growth conditions, it is evident in the second pie chart that the probability of output “low” approaches 50%. The growth conditions have been gradually adjusted to better suit the growth of low-density QDs. It becomes apparent from the third and fourth pie charts that the probability of the model outputting “low” surpasses 55% and persists until the end of the growth.” **on Page 16, Line 280 to Page 16, Line 286 of the manuscript**, and “The output results of the density model are also presented in Figure 5h. Before QDs formation, the density model rarely outputs “high”. However, as the growth process approaches the stage of QD formation, the probability of “high” label output slowly increases to 15%, as demonstrated in the first to third pie charts. Following the formation of QDs, the probability of the density model outputting “high” rapidly increases to 30%, as depicted in the fourth pie chart. Due to the unique nature of QD systems, closing the shutter at this point would restrict the formation of high-density QDs. As the growth process continues and approaches the moment the shutter is closed, it becomes evident that the probability of the model outputting “high” rapidly increases to over 70%, illustrated in the fifth and sixth pie charts.” **on Page 18, Line 317 to Page 19, Line 325 of the manuscript**. Additionally, we have included citations in the manuscript.

Q8: Why did the authors choose ResNet 50 as an ML-based model? There was no comparison between the ResNet 50 model and other ML-based models.

We thank the reviewer for pointing it out. We agree with the reviewer that it is necessary to discuss the comparison between the ResNet 50 model and other ML-based models.

Actually, we have compared the performance among different models using 8 frames

of 300×300 pixels images as model inputs. We have chosen several common network architectures for our research. ResNet 50 and DenseNet are commonly used CNN models for computer vision tasks like image classification and object detection. Our objective is to compare how well these architectures handle residual connections and dense connections. Additionally, we have conducted comparisons with lightweight models. ShuffleNet introduces channel rearrangement operations to curtail computational complexity while preserving performance levels, while MobileNet employs depth-separable convolutions to reduce computational load and model parameters while maintaining commendable performance. These two models have relatively low hardware requirements, and they hold significant promise if they can deliver strong performance.

As illustrated in Fig. R11, the ResNet 50 model exhibited superior performance compared to DenseNet, MobileNet, and ShuffleNet.

Fig. R11: Comparison of training and validation results of different models.

To make it clear, we included the following sentence “We also compared the ResNet 50 model and other models, and the results revealed that ResNet 50 outperforms the other models significantly.” on **Page 11, Line 197** to **Page 11, Line 198** of the manuscript, and S7 titled “Comparison of training and validation results of different models” on **Page 12, Line 176** to **Page 13, Line 184** and Fig. S8 of the supplementary information.

Q9: In Fig. 4, based on the “QDs model”, the QDs were just formed, and the In shutter closed almost at the same time, allowing the authors to collect the “low” QDs density of $3.8 \times 10^8 \text{ cm}^{-2}$. If the target QDs density was larger, for example, $1.7 \times 10^9 \text{ cm}^{-2}$, can their software still judge that it is the ‘low’ label and produce the exact number of the target

QDs? I wonder how precise their system is to produce a specific amount of density.

We have fully addressed the comment.

As density of $1.7 \times 10^9 \text{ cm}^{-2}$ corresponds to the label “low”, the software still judge that it is the “low” label.

Achieving precise density partitioning is the limitations of the current framework and stands for the future direction within the QDs domain. The current label segmentation in the manuscript is based on a relatively broad classification of features that are easily discernible. As the dataset expands and more advanced data recognition methods are introduced in the future, we believe that the current labels in the manuscript can be more finely subdivided, enabling controlled epitaxy for specific quantum dot densities.

To make it clear, we included the following sentence “Moreover, as the breakthroughs in data label partitioning methods and dataset expansion in the future, we anticipate achieving finer-grained density regulation during the material growth process.” to clearly explain the limitations of the current framework on **Page 20, Line 345 to Page 20, Line 346 of the manuscript.**

Q10: In Fig. 4f, despite a significant increase in the probability of the output 'Yes'; the majority of the pie chart represents the output 'No'; Does this mean that the experimental result did not satisfy the 'low' label?

We have fully addressed the comment.

The experimental results have undergone validation through RHEED and AFM, and both align with the “low” label.

We appreciate the Reviewer raised questions about the pie charts we included in the original manuscript; we have updated them according to reviewers’ suggestions to better show the performance of our method.

To effectively depict data trends, we have adjusted pie charts in the manuscript. The statistical analysis conducted on the initial 200 sequences is intended to verify whether the

initial growth conditions are capable of producing QDs with the predetermined target density. The subsequent analysis, spanning from the 200th sequence to 40 sequences before QD formation, serves as a comparative examination against the initial 200 sequences. This stage helps ascertain whether the material growth conditions have been progressively adjusted to better align with the desired growth conditions. The 40 sequences before QD formation are analyzed to demonstrate the QD model’s potential to detect the initiation of QD formation. Finally, an analysis of the 40 sequences after QD formation is conducted to affirm the successful formation of QDs and to portray the ultimate outcome and density change trend of the QDs.

To make it clear, we have updated all pie charts in the manuscript and included the following sentence “To effectively depict data trends, we have incorporated four pie charts throughout to illustrate the evolution of the model’s output results, covering a statistical range spanning the initial 200 sequences, from the 200th sequence to 40 sequences before QD formation, the final 40 sequences before QD formation, and 40 sequences after QD formation.” on **Page 15, Line 267** to **Page 16, Line 270** of the manuscript.

While verifying and modifying the pie chart, we identified errors in the original Fig. 4f in data processing based on the RHEED video. We observed that the formation of QDs took place at the 1600th sequence position. However, the sequence between 1600th and 1640th was not included in the original pie chart, which corresponds to 5 seconds after the shutter was closed. The data after correction is shown in Fig. R12. From the pie chart on the right side of the Fig. R12, it can be clearly seen that the probability of the QD model outputting “Yes” significantly increased from 27% to over 55% before and after the shutdown of the shutter.

Fig. R12: The prediction results of the “QDs model” in the experiment targeting the “low” label, blue line: the QD formation time; yellow line: the In shutter closing time.

To make it clear, we have updated all pie charts in the manuscript and included the following sentence “At the beginning of growth, as shown in the first pie chart in Figure 4f, the primary output of the QDs model was “No”, and the probability of outputting “Yes” was only 7%. In the second pie chart, even though the proportion of outputting “Yes” increases to 15%, the likelihood of the model outputting “No” remains higher. When the growth process approaches the QDs formation, we observed a significant increase from 15% to over 27% in the third pie chart. This trend continues until the shutter closes, as shown in the fourth pie chart with over 55% in the probability of outputting “Yes”. This indicates that the QDs model exhibits a certain sensitivity to the presence of low-density QDs.” on **Page 16, Line 271 to Page 16, Line 278 of the manuscript.**

We have also reviewed all the data processing in the manuscript to assure their accuracy.

Furthermore, we repeated the experiment under identical conditions and obtained low-density samples successfully. As shown in Figure R13a, the substrate temperature underwent a total change of 25 °C during the growth, indicating that the initial substrate temperature was unsuitable for achieving the “low” label target. Furthermore, the convergence of the blue and yellow lines at the same sequence number suggests that, upon QD formation, the “density model” promptly recognized the current RHEED sequence as consistent with the “low” label and terminated the growth process. In Figure R13b, RHEED images primarily displayed streaks during the initial growth stage, even before QD formation, as shown in Figure R13c. After QD formation, distinctive streaks and spot features promptly emerged, aligning with the characteristics of the “low” label, as depicted in Figure R13d. The density of these QDs was $3.7 \times 10^8 \text{ cm}^{-2}$, with an average diameter of 44.8 nm and a height of 7.3 nm, as evidenced in the AFM image in Figure R13e.

Figures R13f and R13g illustrate the distribution of outputs for the QDs and density models across the entire sequence. To effectively depict data trends, we have incorporated multiple pie charts to illustrate the model’s output evolution. Consequently, each model’s

scatter plot now includes four pie charts, covering a statistical range spanning the initial 200 sequences, from the 200th sequence to 40 sequences before QD formation, the final 40 sequences before QD formation, and 40 sequences after QD formation. Before QDs formation, the model predominantly outputs “No”, as depicted in the first and second pie charts of Figure R13f. However, as the sequence approaches the blue line, the likelihood of observing “Yes” significantly rises and surpasses 55%, as shown in the third pie chart. As illustrated in the fourth pie chart, following the QDs formation, the probability of the QDs model outputting “Yes” further increases to 65%. It indicates the model’s sensitivity in detecting the emergence of low-density QDs.

Moreover, Figure R13g presents the results of the density model. From the first pie chart to the second pie chart, we can observe a gradual increase in the probability of the density model outputting “low”. This trend continues until the third and fourth pie charts, with the probability of the “low” label remaining stable at around 60%. This trend confirms that the model continuously adjusts growth parameters to create more favorable growth conditions for low-density QDs.

Fig. R13: Additional experiment on controlling growth process of low-density QDs. Experiment with the “low” label as the target. (a) Substrate temperature changes during growth. The RHEED image (b) captured at 200th frame after growth; (c) before the QD formation; (d) after the QD formation. (e) the 2 μm × 2 μm AFM image of the sample; the prediction results of (f) the “QDs model” and (g) the “density model”.

The additional experimental data are also included in S13 titled “Another controlled growth experiment of low-density QDs” on **Page 18, Line 246** to **Page 19, Line 272** and Fig. S15 of the **supplementary information**.

Q11: In Fig. 5, based on the “QDs model”, the authors said that “It is noteworthy that there is still a distance between the blue line and the last “No” label, since we confirm the QD formation by collecting multiple results of the model”. Does it mean that their

model is not accurate enough?

We apologize for the misunderstanding. In our research, the average validation accuracy of the density model reached 95.1%.

We collected multiple results of the model not because the model is not accurate enough. Our approach involves integrating the output of a single model to reduce the impact of randomness and variability while focusing on the functionality of a single model. (Wolpert D H. Stacked generalization[J]. Neural networks, 1992, 5(2): 241-259; Kuncheva L I. Combining pattern classifiers: methods and algorithms. John Wiley & Sons, 2014) (Dietterich T G. Ensemble methods in machine learning. International workshop on multiple classifier systems. Berlin, Heidelberg: Springer Berlin Heidelberg, 2000: 1-15) This can reduce the likelihood of operational errors in the model during software deployment.

To make it clear, we included the following sentence, “Furthermore, we collect multiple results from the QDs and density models outputs when deploying. The result with the highest probability is selected to determine whether the adjustment to the growth parameters is necessary. This approach helps mitigate the impact of randomness and variability, reducing the likelihood of operational errors by the model during software deployment. (Wolpert D H. Stacked generalization[J]. Neural networks, 1992, 5(2): 241-259; Kuncheva L I. Combining pattern classifiers: methods and algorithms. John Wiley & Sons, 2014; Dietterich T G. Ensemble methods in machine learning. International workshop on multiple classifier systems. Berlin, Heidelberg: Springer Berlin Heidelberg, 2000: 1-15)” on **Page 12, Line 221 to Page 12, Line 225 of the manuscript.**

The mentioned references are also cited in the manuscript.

Q12: In Fig. 5, the output ‘Yes’ in the ‘QDs model’ started at a sequence around 1050, but why did the In shutter close at a sequence around 1400? If the In shutter had closed right after the QDs formation, could ‘high’ density QDs have been collected? I doubt the feasibility of their model to predict the QD formation.

We have fully addressed the comment.

QD growth is a phased development process, particularly from the initial formation of QDs to the attainment of high-density QDs, which requires a significant amount of time. Closing the shutter immediately will prevent the formation of high-density samples. During the initial stages of InAs quantum dot formation, the distribution of InAs atoms on the substrate surface is relatively scattered, and the distance between QDs is relatively wide. Closing the shutter too early in this phase results in an insufficient supply of InAs atoms, preventing the continued growth of existing QDs or the formation of new ones. Consequently, this limitation restricts the increase in both the density and size of quantum dots, making it impossible to achieve high-density quantum dot arrays (<https://doi.org/10.1088/0953-8984/19/22/225006>)(<https://doi.org/10.1063/1.116580>).

To make it clear, we included the following sentence, “Due to the unique nature of QD systems, closing the shutter at this point would restrict the formation of high-density QDs. (Placidi E, Arciprete F, Fanfoni M, et al. InAs/GaAs (001) epitaxy: kinetic effects in the two-dimensional to three-dimensional transition. *Journal of Physics: Condensed Matter*, 2007, 19(22): 225006; Kobayashi N P, Ramachandran T R, Chen P, et al. In situ, atomic force microscope studies of the evolution of InAs three-dimensional islands on GaAs (001). *Applied physics letters*, 1996, 68(23): 3299-3301).” on **Page 19, Line 321** to **Page 19, Line 323** of the manuscript.

The mentioned reports are also cited in the manuscript.

Q13: In Fig. S9, the ‘density model’ prediction results showed an increase in the ‘high’ label as the target. However, the majority of the results still belonged to the ‘low’ label, and it did not fit with the target label.

We appreciate the Reviewer raised questions about the pie charts we included in the original manuscript; we have updated them according to reviewers’ suggestions to better show the performance of our method. We have improved the pie chart and relocated it to

Fig. 5h of the manuscript.

To make it clear, we have updated all pie charts in the manuscript and included the following sentence “In high-density growth experiments, the shutter remains open after the formation of QDs for a period. Therefore, we incorporated two additional pie charts: one for the 40 sequences before the shutter closure and another for the 40 sequences after the shutter closure.” on **Page 18, Line 307 to Page 1, Line 309 of the manuscript.**

In original Fig. S9, most of the density model’s final output results are labeled as “high” and correspond to AFM result. By adding separate pie charts before and after closing the shutter, we observed that, compared to the pie chart results before closure, the “high” label probability in the pie chart exceeded 70%, significantly higher than other labels, as shown in Fig. R5.

To make it clear, we have provided additional descriptions for the pie charts to effectively illustrate the trends in data changes, “As the growth process continues and approaches the moment the shutter is closed, it becomes evident that the probability of the model outputting “high” rapidly increases to over 70%, illustrated in the fifth and sixth pie charts.” on **Page 19, Line 323 to Page 19, Line 325 of the manuscript.** Additionally, we have included this citation in the manuscript.

Reviewer 4:

Q1: In the example optimizing towards a low surface density of QDs, the model never reliably predicts quantum dot formation during the optimization based on the contents of Figure 4f, even in the later epochs it is still predicting "no" QD formation in ~80% of cases. Similarly, the density prediction seems to not improve particularly over the course of the optimization either (Figure S8). In the example of the high surface density optimization, where the signal is better the model does apparently accurately predict QD formation, although the density of the QDs synthesized (Figure 5f) are much lower than the target densities (Figure 1e&f). Figure S9 seems to suggest that the optimization never reliably finds conditions suitable to produce the high density films. Additionally, I think the article

could do with reorganization before publication in any journal, there are details in the main text about the ML model, which are too technical for the main text (but are important and should be included in the SI) and results that are important such as the density prediction model, which are only in the SI.

We thank the reviewer for raising the question on the performance of the models and suggesting how to improve this article. Our responses are as follows:

(A) In the example optimizing towards a low surface density of QDs, the model never reliably predicts quantum dot formation during the optimization based on the contents of Figure 4f, even in the later epochs it is still predicting "no" QD formation in ~80% of cases.

We appreciate the Reviewer raised questions about the pie charts we included in the original manuscript; we have updated them according to reviewers' suggestions to better show the performance of our method.

To effectively depict data trends, we have adjusted pie charts in the manuscript. The statistical analysis conducted on the initial 200 sequences is intended to verify whether the initial growth conditions are capable of producing QDs with the predetermined target density. The subsequent analysis, spanning from the 200th sequence to 40 sequences before QD formation, serves as a comparative examination against the initial 200 sequences. This stage helps ascertain whether the material growth conditions have been progressively adjusted to better align with the desired growth conditions. The 40 sequences before QD formation are analyzed to demonstrate the QD model's potential to detect the initiation of QD formation. Finally, an analysis of the 40 sequences after QD formation is conducted to affirm the successful formation of QDs and to portray the ultimate outcome and density change trend of the QDs.

To make it clear, we have updated all pie charts in the manuscript and included the following sentence "To effectively depict data trends, we have incorporated four pie charts throughout to illustrate the evolution of the model's output results, covering a statistical

range spanning the initial 200 sequences, from the 200th sequence to 40 sequences before QD formation, the final 40 sequences before QD formation, and 40 sequences after QD formation.” on **Page 15, Line 267** to **Page 16, Line 270** of the manuscript.

While verifying and modifying the pie chart, we identified errors in the original Fig. 4f in data processing based on the RHEED video. We observed that the formation of QDs took place at the 1600th sequence position. However, the sequence between 1600th and 1640th was not included in the original pie chart, which corresponds to 5 seconds after the shutter was closed. The data after correction is shown in Fig. R14. From the pie chart on the right side of the Fig. R14, it can be clearly seen that the probability of the QD model outputting “yes” significantly from 27% to over 55% before and after the shutdown of the shuttle.

Fig. R14: The prediction results of the “QDs model” in the experiment targeting the “low” label, blue line: the QD formation time; yellow line: the In shutter closing time.

To make it clear, we have updated all pie charts in the manuscript and included the following sentence “At the beginning of growth, as shown in the first pie chart in Figure 4f, the primary output of the QDs model was “No”, and the probability of outputting “Yes” was only 7%. In the second pie chart, even though the proportion of outputting “Yes” increases to 15%, the likelihood of the model outputting “No” remains higher. When the growth process approaches the QDs formation, we observed a significant increase from 15% to over 27% in the third pie chart. This trend continues until the shutter closes, as shown in the fourth pie chart with over 55% in the probability of outputting “Yes”. This indicates that the QDs model exhibits a certain sensitivity to the presence of low-density QDs.” on **Page 16, Line 271** to **Page 16, Line 278** of the manuscript.

We have also reviewed all the data processing in the manuscript to assure their

accuracy.

Furthermore, we repeated the experiment under identical conditions and obtained low-density samples successfully. As shown in Figure R15a, the substrate temperature underwent a total change of 25 °C during the growth, indicating that the initial substrate temperature was unsuitable for achieving the “low” label target. Furthermore, the convergence of the blue and yellow lines at the same sequence number suggests that, upon QD formation, the “density model” promptly recognized the current RHEED sequence as consistent with the “low” label and terminated the growth process. In Figure R15b, RHEED images primarily displayed streaks during the initial growth stage, even before QD formation, as shown in Figure R15c. After QD formation, distinctive streaks and spot features promptly emerged, aligning with the characteristics of the “low” label, as depicted in Figure R15d. The density of these QDs was $3.7 \times 10^8 \text{ cm}^{-2}$, with an average diameter of 44.8 nm and a height of 7.3 nm, as evidenced in the AFM image in Figure R13e.

Figures R15f and R15g illustrate the distribution of outputs for the QDs and density models across the entire sequence. To effectively depict data trends, we have incorporated multiple pie charts to illustrate the model’s output evolution. Consequently, each model’s scatter plot now includes four pie charts, covering a statistical range spanning the initial 200 sequences, from the 200th sequence to 40 sequences before QD formation, the final 40 sequences before QD formation, and 40 sequences after QD formation. Before QDs formation, the model predominantly outputs “No”, as depicted in the first and second pie charts of Figure R15f. However, as the sequence approaches the blue line, the likelihood of observing “Yes” significantly rises and surpasses 55%, as shown in the third pie chart. As illustrated in the fourth pie chart, following the QDs formation, the probability of the QDs model outputting “Yes” further increases to 65%. It indicates the model’s sensitivity in detecting the emergence of low-density QDs.

Moreover, Figure R15g presents the results of the density model. From the first pie chart to the second pie chart, we can observe a gradual increase in the probability of the density model outputting “low”. This trend continues until the third and fourth pie charts, with the probability of the “low” label remaining stable at around 60%. This trend confirms

that the model continuously adjusts growth parameters to create more favorable growth conditions for low-density QDs.

Fig. R15: Additional experiment on controlling growth process of low-density QDs. Experiment with the “low” label as the target. (a) Substrate temperature changes during growth. The RHEED image (b) captured at 200th frame after growth; (c) before the QD formation; (d) after the QD formation. (e) The 2 μm × 2 μm AFM image of the sample; the prediction results of (f) the “QDs model” and (g) the “density model”.

The additional experimental data are also included in 13 titled “Another controlled growth experiment of low-density QDs” on Page 18, Line 246 to Page 19, Line 272 and Fig. S15 of the supplementary information.

(B) Similarly, the density prediction seems to not improve particularly over the course

of the optimization either (Figure S8).

In fact, the density model results have shown gradual optimization.

Using the pie chart method described above, the modified original Fig. S8 is displayed as Fig. R16. When examining the proportion of “low” labels in the four pie charts, it becomes clear that as the sequence advances, the likelihood of the density model outputting “low” gradually rises. This indicates that the gradual adjustment of growth conditions is more favorable for the growth of low-density QDs.

Fig. R16. The prediction results of the “density model” in the experiment targeting the “low” label, blue line: the QD formation time; yellow line: the In shutter closing time.

To make it clear, we included the following sentence, “Figure 4g shows the “density model” results of the experiment with the “low” label as the target. In the first pie chart, the probability of output “low” is significantly lower than that of combined “middle” and “high”. However, with the real-time feedback control of growth conditions, it is evident in the second pie chart that the probability of output “low” approaches 50%. The growth conditions have been gradually adjusted to better suit the growth of low-density QDs. It becomes apparent from the third and fourth pie charts that the probability of the model outputting “low” surpasses 55% and persists until the end of the growth.” on **Page 16, Line 280** to **Page 16, Line 286** of the manuscript. Additionally, we have incorporated this citation into the manuscript and have relocated original Fig. S8 of the supplementary information to Fig. 4g of the manuscript.

(C) In the example of the high surface density optimization, where the signal is better

the model does apparently accurately predict QD formation, although the density of the QDs synthesized (Figure 5f) are much lower than the target densities (Figure 1e&f).

Apologies for any inconvenience caused. The QD density in Fig. 1f is $1.0 \times 10^{11} \text{ cm}^{-2}$, whereas the QD density in Fig. 5f is higher at $1.4 \times 10^{11} \text{ cm}^{-2}$. As the density of QDs increases, the size of QDs decreases, resulting in more dense dots in AFM images.

(D) Figure S9 seems to suggest that the optimization never reliably finds conditions suitable to produce the high density films.

We appreciate the Reviewer for pointing out the issue. We totally agree that the original Fig. S9 and other similar figures need further improvement to show the results.

To effectively illustrate the changes in data trends, we have included an additional analysis of the results obtained from the density models for the 40 sequences both before and after the closure of the shutter, as shown in Fig. R17. The probability of the density model outputting a “high” label experiences a substantial increase of over 70% before and after the shutter is closed.

Fig. R17. The prediction results of the “density model” in the experiment targeting the “high” label, blue line: the QD formation time; yellow line: the In shutter closing time.

To make it clear, we included the following sentence, “In high-density growth experiments, the shutter remains open after the formation of QDs for a period. Therefore, we incorporated two additional pie charts: one for the 40 sequences before the shutter closure and another for the 40 sequences after the shutter closure.” on **Page 18, Line 307**

to **Page 18, Line 309** and Fig. 5h of **the manuscript**, “The output results of the density model are also presented in Figure 5h. Before QDs formation, the density model rarely outputs “high”. However, as the growth process approaches the stage of QD formation, the probability of “high” label output slowly increases to 15%, as demonstrated in the first to third pie charts. Following the formation of QDs, the probability of the density model outputting “high” rapidly increases to 30%, as depicted in the fourth pie chart. Due to the unique nature of QD systems, closing the shutter at this point would restrict the formation of high-density QDs. As the growth process continues and approaches the moment the shutter is closed, it becomes evident that the probability of the model outputting “high” rapidly increases to over 70%, illustrated in the fifth and sixth pie charts.” on **Page 18, Line 317 to Page 19, Line 325 of the manuscript**.

(E) **Additionally, I think the article could do with reorganization before publication in any journal, there are details in the main text about the ML model, which are too technical for the main text (but are important and should be included in the SI) and results that are important such as the density prediction model, which are only in the SI.**

We have fully addressed the comment.

We have moved the details regarding the parameters and structures of the ResNet 50 model from the manuscript to S5 titled “The detailed structure of the ResNet 50 model” of the supplementary information and moved the original Fig. S8 and S9 concerning density mode results from the supplementary information to Fig. 4g and 5h of the manuscript.

Q2:

Abstract

***RHEED is not defined.**

***The phrase "specifically designed for training RHEED videos" does not make sense. The ML model is trained 'using' RHEED videos as an input**

***"In situ" should not be hyphenated**

We have fully addressed the comment.

We have defined RHEED by reflection high-energy electron diffraction (RHEED).

We totally agree that the phrase “specifically designed for training RHEED videos” is not appropriate. We have changed the phrase to “trained using reflection high-energy electron diffraction (RHEED) videos as input”.

The hyphens for in situ have been removed for the abstract and the manuscript.

To make it clear, we included the following sentence, “We developed a machine learning (ML) model named 3D ResNet 50 trained using reflection high-energy electron diffraction (RHEED) videos as input instead of static images and providing real-time feedback on surface morphologies for process control.” on **Page 3, Line 41** to **Page 3, Line 43** of the manuscript.

Q3:

Line 52

* I don't agree that ML is capable of "establishing complex systems' empirical functions". ML regressors are capable of representing these underlying relationships within a dataset through what is essentially (potentially very high-dimensional) curve-fitting. Obtaining such functions requires an inverse design approach from a well-trained model.

We totally agree with the comments on the capability of ML.

We believe that ML regressors can, to some extent, capture the empirical functions of complex systems by learning patterns in data and performing curve fitting. However, I fully concur with the significance of the inverse design method you mentioned. In certain situations, particularly when dealing with intricate systems, it may be necessary to integrate ML models with inverse design methods to deduce the empirical functions of the system.

To make it clear, we included the following sentence, “Machine learning (ML) is revolutionary due to its exceptional capability for pattern recognition and its potential in approximating the empirical functions of complex systems.” on **Page 4, Line 67** to **Page 4,**

Line 68 of the manuscript.

Q4:

Line 57

* "once the sample characterization results deviate" - deviate from what, please specify

We have fully addressed the comment.

We have added “from expectation” after “deviate”.

To make it clear, we included the following sentence, “Therefore, once the sample characterization results deviate from expectation, it becomes challenging and time-consuming to identify reasons.” on **Page 5, Line 79** to **Page 5, Line 80** of the manuscript.

Q5:

Line 69

* "growers" - would "MBE users" be a better phrasing?

In this field, we normally call researchers who use MBE to grow materials as “growers”. Therefore, we wish to keep the phrasing “growers”.

Q6:

Line 80-81

* for better readability move "automatically and intelligently" before the word achieving.

We agree with the reviewer that it has better readability if we moved “automatically and intelligently” before the word achieving.

To make it clear, we included the following sentence, “In this work, we proposed a real-time feedback control based on ML to connect in situ RHEED videos and QD density, automatically and intelligently achieving a low-cost, efficient, reliable, and reproduceable growth of QDs with required density.” on **Page 5, Line 96** to **Page 5, Line 98** of the

manuscript.

Q7:

Line 100-2

* the phrasing here is confusing. It needs to be specified that the authors used AFM to determine that the surface was flat and the label "zero" refers to the density of QDs on the surface.

We have fully addressed the comment.

We changed the sentence to “The samples grown without QDs have flat surfaces determined by an atomic force microscope (AFM) as shown in Figures 1c, corresponding to streaky RHEED patterns in Figure 1g, which was artificially labeled with “zero” referring to the density of QDs on the surface.” on **Page 6, Line 119** to **Page 6, Line 122** of **the manuscript**.

Q8:

Line 109

* Remove the word "exceeding", as this definition includes the "high" condition

We sincerely apologize for any ambiguity in our manuscript. To address this, we have updated the definition of different density levels.

To make it clear, we included the following sentence, “Upon conducting a more thorough analysis of the RHEED patterns, we identified the patterns transition from having both streaks and spots to only spots, corresponding to a density of $1 \times 10^{10} \text{ cm}^{-2}$ in AFM images. So, we assigned the “low” label to densities below $1 \times 10^{10} \text{ cm}^{-2}$, the “middle” label to densities ranging from $1 \times 10^{10} \text{ cm}^{-2}$ to $4 \times 10^{10} \text{ cm}^{-2}$. As the density exceeds $4 \times 10^{10} \text{ cm}^{-2}$, the spot features become more rounded and show higher brightness, we labeled densities exceeding $4 \times 10^{10} \text{ cm}^{-2}$ as “high”.” on **Page 7, Line 127** to **Page 7, Line 132** of **the manuscript**.

Q9:

Figure 1c-f, 4e,S7

- * The labels for each condition are not given on the Figure or in the caption.
- * please provide a scale bar for the heights in the AFM images. Without this the images do not support the claims in the text about flatness. Ideally the Z-scale should be the same for all images that are being compared, which currently it is not.

We thank the reviewer for the suggestion.

We have included labels for each image and incorporated a scale bar into all AFM images. Furthermore, the Z-scale of all images is now the same, which spans from -5 nm to +5 nm.

To make it clear, these can be found in Fig. 1c-f, 4e, 5f of **the manuscript**, and Fig. S14a, S15e, S16e of **the supplementary information**.

Q10:

Figure 1g-j

- * The streaks in the RHEED are not very visible when the article is printed out on paper, is it possible to alter the contrast of the images without comprising the data?

We thank the reviewer for pointing out the quality issue for these images.

We have modified the image display method in Fig. 1g-j of **the manuscript** to improve contrast and clarity.

Q11:

Figure 2a

- * Personally, I think the representation of the ML model is not great. I think that it would be easier for many readers if the authors clearly represented the inputs and the outputs of the model, rather than the images of the stacked grids and the parts of the MBE with no

explanation of what they are supposed to represent. Something like the top part of Figure 6 of this paper, for me is much clearer and useful to the reader. <https://onlinelibrary.wiley.com/doi/full/10.1002/cjoc.202000393> (this is not a request to cite this paper, it is just the first example that I found of a suitable diagram).

We appreciate reviewer's suggestion concerning the need for a more concise and intuitive chart to represent the model's functionality. The example provided by the reviewer indeed helps improve the representation of the ML model.

The Fig. 2a of the original manuscript serves the purpose of illustrating the comprehensive design framework, encompassing more than just the model itself. The original Fig. 3 has been moved to S5 of the supplementary information, and we have replaced Fig. 3 in the manuscript by adopting the format of the top part of Fig. 6 of the paper mentioned by the Reviewer. At the same time, we have also added a control logic diagram here to demonstrate how the model is used.

As depicted in Fig. R18a, the diagram provides a brief overview of the model's framework and functionalities, with different outputs being generated using two models.

Fig. R18b illustrates the control logic we implemented in the LabVIEW program. Upon opening the shutter, if the label output from the density model aligns with the preset target before QD formation, the current substrate temperature remains unchanged. If the density model's output label exceeds the predefined target density, the substrate temperature increases; otherwise, it decreases. Once the QDs model recognizes QD formation, the shuttle will remain open until the density model's output matches the preset target. At that point, it closes the shutter to complete the growth process. This control logic continually adjusts the substrate temperature, gradually increasing the likelihood of growing QDs with the desired target density.

Fig R18. (a) A simplified architectural diagram of the model; (b) a control logic diagram for the model deployment.

To make it clear, we included the following sentence, “As shown in Figure 3a, the output of the QDs model consists of just two categories, represented as “Yes” or “No”, indicating whether the QD has formed. In contrast, the output of the density model consists of “zero”, “low”, “middle”, and “high” labels, corresponding to different QD density prediction results.” on **Page 11, Line 203** to **Page 12, Line 206** and Fig. 3a of **the manuscript**.

“After opening the shutter, the substrate temperature remains unchanged if the label output from the density model aligns with the preset target before QD formation. If the output label’s density from the density model exceeds the target density, the substrate temperature increases; otherwise, it decreases. Once the QDs model recognizes QD formation, the shuttle will remain open until the density model’s output matches the target. At this point, it closes the shutter to complete the growth process. This control logic adjusts the substrate temperature continuously, gradually increasing the likelihood of growing QDs with the target density.” on **Page 13, Line 233** to **Page 13, Line 241** and Fig. 3d of **the**

manuscript.

The mentioned references are also cited in the manuscript.

Q12:

Figure 2c

*this Figure is redundant, the cropped region could easily be indicated in part b

We agree with the reviewer this Figure is not necessary.

We have integrated the original Fig. 2c into the original Fig. 2b and have updated the figure captions and corresponding text descriptions.

To make it clear, these can be found in Fig. 2b of **the manuscript**.

Q13:

Line 155

* The Authors should introduce that 3D ResNet 50 Model is the model that they are using at the start of this paragraph, and briefly describe what it does/why it was used, rather than jumping straight into details about which residual blocks they used.

We agree with the reviewer it will be much better if we introduce that 3D ResNet 50 Model is the model that we are using with details at the start of this paragraph.

We explained our selection of ResNet 50 by highlighting its advantages, including fast convergence, mitigation of gradient vanishing problems, and suitability for deep training. Furthermore, we conducted a detailed comparison between our model and the standard ResNet model.

To make it clear, these can be found in “The ResNet 50 model has emerged as our preferred choice for several reasons, including its capability to address gradient vanishing issues and automatically learn methods for extracting features from raw data without manual feature engineering or dimensionality reduction, rapid convergence, and suitability for deep training.” on **Page 11, Line 183** to **Page 11, Line 186** of **the manuscript**, and “We

also compared the ResNet 50 model and other models, and the results revealed that ResNet 50 outperforms the other models significantly.” on **Page 11, Line 197** to **Page 11, Line 198** of **the manuscript**.

Q14:

Lines 155-182 and Figure 3

* Following on from this point. The details given in these 2 paragraphs are important, but I feel they are inappropriate in the main text, and would be better placed in the SI for interested readers. I think the level of detail is too high for most readers who will be interested in the implications of this approach on producing materials, rather than the specifics of how the authors set up their CNN. I think that it would be better if this section was replaced with more general explanation of how the machine learning was used. It would also be beneficial to explicitly state which optimization technique you are using here and in the Methods section.

***Why is there no more discussion of the surface density prediction model in the main text after this point? Why is it all hidden in the SI?

We thank the reviewer for suggesting the way to improve our article. Our responses are as follows:

(A) * Following on from this point. The details given in these 2 paragraphs are important, but I feel they are inappropriate in the main text, and would be better placed in the SI for interested readers. I think the level of detail is too high for most readers who will be interested in the implications of this approach on producing materials, rather than the specifics of how the authors set up their CNN. I think that it would be better if this section was replaced with more general explanation of how the machine learning was used.

Thank you for the suggestion. We totally agree that most readers who will be interested

in the implications of this approach on producing materials, rather than the specifics of how to set up CNN.

To make it clear, we have moved the detailed information about the ResNet 50 model and the original Fig. 3a from the manuscript to the supplementary information. These can be found in S5 titled “The detailed structure of the ResNet 50 model” on **Page 8, Line 100** to **Page 11, Line 144** and Fig. S7 of **the supplementary information**. We have explained here how the machine learning was used on **Page 13, Line 233** to **Page 13, Line 241** of **the manuscript**.

(B) **It would also be beneficial to explicitly state which optimization technique you are using here and in the Methods section.**

We have also provided a description of the model optimization techniques we adopted here. This includes a comparison of various candidate models, optimization techniques for the convolutional layer and residual block, as well as specific parameter differences and advantages of our model in comparison to the standard model.

To make it clear, these can be found in “When processing video frame data, 3D convolution proves more effective in extracting spatial and temporal dimension information from videos than the 2D convolution used in the standard ResNet model. The difference between our network and the original ResNet lies in the dimensionality of the convolutional kernel and batch normalization operations. Our 3D ResNet 50 model employs 3D convolutions and 3D batch normalization. This spatiotemporal-feature-learning method enables the 3D ResNet 50 model to better capture the dynamic information in videos compared to 2D CNN, resulting in an enhanced ability to identify and differentiate the categories of videos. Furthermore, we have adjusted the structure of identity shortcuts to reduce information loss during downsampling and reduced the frequency at which the model doubles the number of channels after passing through the residual structure.” on **Page 11, Line 186** to **Page 11, Line 195** of **the manuscript**.

(C) ***Why is there no more discussion of the surface density prediction model in the main text after this point? Why is it all hidden in the SI?

Thank the Reviewer for the suggestions, we totally agree a rewriting of this part is necessary.

We have included a description of the accuracy of the density model in the main text and have conducted an analysis to elucidate the reasons behind fluctuations in the density model data and the challenges associated with further improvements. Regarding the performance of the density model, we have improved the pie chart display method in original Fig. S8 and S9 and relocated them to Fig. 4g and 5h, respectively. Additionally, we have provided descriptions for the pie charts to effectively illustrate the trends in data changes.

To make it clear, we included the following sentence, “Throughout the model training process, the persistent fluctuations and plateaus in the “QDs model” and “density model” accuracy were attributed to the data diversity and the manual label assignment process instead of limitations inherent in the model itself. After 120 training cycles, the “QDs model” maintained a validation accuracy of over 90%, as shown in Figure 3b, ultimately achieving a final average validation accuracy of 94.4%. The fluctuations observed in the “QDs model” validation process predominantly originate from a specific time before the QD formation when the RHEED streaks still dominated most of the collected images. As the QDs are about to form, the spot features closely associated with QD formation are often faint and overshadowed by the streak features. As depicted in Figure 3c, the validation accuracy of the “density model” continues to exhibit fluctuations even after 100 epochs, primarily attributed to label overlap when low-density QD data exhibit an increase in density with deposition. (Tatebayashi J, Nishioka M, Someya T, et al. Area-controlled growth of InAs quantum dots and improvement of density and size distribution. Applied Physics Letters, 2000, 77(21): 3382-3384) However, it consistently maintained a validation accuracy of over 80%, and the final average validation accuracy reached 95.1%.” on **Page 12, Line 208** to **Page 12, Line 220** of the manuscript.

The mentioned references are also cited in the manuscript.

Q15:

Line 224.

* The phrase "significant increase" needs to be quantified. It appears from the pie charts to go from about 15 to about 20 %, which is not so great, if the model is supposed to be accurately predicting QD formation.

We totally agree the phrase “significant increase” needs to be quantified.

To effectively depict data trends, we have adjusted pie charts in the manuscript. While verifying and modifying the pie chart, we identified errors in the original Fig. 4f in data processing based on the RHEED video. We observed that the formation of QDs took place at the 1600th sequence position. However, the sequence between 1600th and 1640th was not included in the original pie chart, which corresponds to 5 seconds after the shutter was closed. The data after correction is shown in Fig. R14. From the first pie chart on the left side of Fig. R14, at the beginning of growth, the probability of QDs model outputting “Yes” is only 7%. As the growth process progresses, it can be observed in the second pie chart that the probability of the QDs model output “Yes” increases to 15%, but it is still extremely low. When the material growth approaches the formation of QDs, we can clearly observe that the probability of the model outputting “Yes” significantly increased from 27% to more than 55%.

To make it clear, we have updated all pie charts in the manuscript and included the following sentence “At the beginning of growth, as shown in the first pie chart in Figure 4f, the primary output of the QDs model was “No”, and the probability of outputting “Yes” was only 7%. In the second pie chart, even though the proportion of outputting “Yes” increases to 15%, the likelihood of the model outputting “No” remains higher. When the growth process approaches the QDs formation, we observed a significant increase from 15% to over 27% in the third pie chart. This trend continues until the shutter closes, as shown in the fourth pie chart with over 55% in the probability of outputting “Yes”. This indicates that the QDs model exhibits a certain sensitivity to the presence of low-density QDs.” on

Page 16, Line 271 to Page 16, Line 278 of the manuscript.

Q16:

Figure 4

* It's hard to resolve the RHEED images when printed out on paper again. If possible it would be better to show a RHEED scattering pattern close to the threshold indicated by the blue line. From the data shown you could only conclude that QD formation occurs somewhere between epoch ~1430 and ~1600

* Finally 4f does not really display very well the number of "yes" and "no" conditions, it seems like "no" happens constantly throughout all epochs because the points overlap each other. I think this format is very poor in almost all instances that it is used, adding a running average across all epochs would demonstrate the changes more clearly to the reader, if there is a major change.

It is also unclear which portions of the graph the two inset pie charts apply to. Is the left one all epochs below the blue line and the right all after?, this needs to be made clearer. This is true for the other instances of pie charts in the main text and SI also.

We thank the reviewer for suggesting the way to improve our article. Our responses are as follows:

(A)* It's hard to resolve the RHEED images when printed out on paper again.

We have updated the images and made adjustments to the brightness and contrast to enhance the clarity of the RHEED pattern, as shown in Fig. R19.

Fig. R19. Experiment with the “low” label as the target. The RHEED image (a) captured at 200th frame after growth; (b) before the QD formation; (c) after the QD formation.

To make it clear, these can be found in Fig. 4b-d of **the manuscript**.

(B) If possible it would be better to show a RHEED scattering pattern close to the threshold indicated by the blue line. From the data shown you could only conclude that QD formation occurs somewhere between epoch ~1430 and ~1600

We added a RHEED scattering pattern close to the threshold indicated by the blue line, to replace the original Fig. 4c.

To make it clear, these can be found in Fig. 4a and 4c of **the manuscript**.

(C)* Finally 4f does not really display very well the number of "yes" and "no" conditions, it seems like "no" happens constantly throughout all epochs because the points overlap each other. I think this format is very poor in almost all instances that it is used, adding a running average across all epochs would demonstrate the changes more clearly to the reader, if there is a major change.

We really appreciate the reviewer’s suggestion to improve our manuscript.

We attempted to substitute the scatter plot with the running average values across all sequence. Fig. R20 illustrates that the transformed data exhibits a curve-like pattern. However, since it consists of labeled data rather than numerical values, we can discern the curve’s variation pattern but cannot precisely define the average value for label conversions.

Fig. R20. The QDs model results (a) and density model results (b) are presented as running average values across all sequences.

Therefore, we have adjusted pie charts in the manuscript to effectively depict data trends. The original Fig. 4f and Fig. S8 have been transformed and are now presented in Fig. R14 and Fig. R16.

We have also updated all pie charts in the manuscript to demonstrate the changes more clearly to the readers.

To make it clear, these can be found in Fig. 4f-4g, 5g-5h of **the manuscript**, and Fig. S15f-g, S16f-g of **the supplementary information**.

(D)It is also unclear which portions of the graph the two inset pie charts apply to. Is the left one all epochs below the blue line and the right all after?, this needs to be made clearer. This is true for the other instances of pie charts in the main text and SI also.

Before introducing the pie chart for the first time in the manuscript, we explicitly defined the data range for each pie chart. Furthermore, we provided a separate explanation of the additional statistical data range for the pie chart in the description of Fig. 5 in the

manuscript.

To make it clear, we have updated all pie charts in the manuscript and included the following sentence “To effectively depict data trends, we have incorporated four pie charts throughout to illustrate the evolution of the model’s output results, covering a statistical range spanning the initial 200 sequences, from the 200th sequence to 40 sequences before QD formation, the final 40 sequences before QD formation, and 40 sequences after QD formation.” on **Page 15, Line 267** to **Page 16, Line 270** of the manuscript.

“In high-density growth experiments, the shutter remains open after the formation of QDs for a period. Therefore, we incorporated two additional pie charts: one for the 40 sequences before the shutter closure and another for the 40 sequences after the shutter closure.” on **Page 18, Line 307** to **Page 1, Line 309** of the manuscript.

Q17:

Line 233-4

* The sentence defining what "yes" or "no" means in relation to the ML model should appear earlier in the paragraph. Preferably on line 223, before the phrase "Before the QD formation..."

We have fully addressed the comment.

We have relocated the sentence to just before the phrase “Before the QD formation...”

To make it clear, we included the following sentence, “We classified RHEED videos as the “Yes” or “No” based on observations of QDs formation.” on **Page 16, Line 270** to **Page 16, Line 271** of the manuscript.

Q18:

Figure 4f

* The AFM image does not look particularly dense compared to the examples of "high density" in Figure 1

We believe that the Reviewer was mentioning Fig. 5f instead of 4f.

We apologize for any inconvenience caused by the absence of detailed labels on the original AFM image. We have now updated all AFMs throughout the text and included labels for each of them. The label corresponding to original Fig. 5f was “high”.

To make it clear, these can be found in Fig. 1c-f, 4e, 5f of **the manuscript**, and Fig. S14a, S15e, S16e of **the supplementary information**.

Q19:

Figure 4g

* It is not clear to me why the model starts predicting QD formation before the blue line

We have fully addressed the comment.

The key goal of this research is to intervene in the material growth process before the QD formation (blue line) assisted by the model prediction, which will increase the possibility of the achieving the predetermined material. This is also the major advance of proposed approach over existing methods. Therefore, we must be involved in actively and adjust the material’s growth parameters continuously, from the initial stage of growth to QD formation and continues until the end of growth. This approach enables us to achieve real-time feedback control of the QD growth process.

To make it clear, we included the following sentence, “To enable early intervention in the growth process before the QDs formation, we developed a control logic for the LabVIEW program, as shown in Figure 3d.” on **Page 13, Line 233** to **Page 13, Line 234** of **the manuscript**.

Q20:

Line 304

* STAIB is not defined

STAIB is a company name, we changed to its full name “STAIB Instruments” to make

it clear.

To make it clear, these can be found in “RHEED in the MBE growth chamber enabled us to analyze and monitor the surface of the epilayer during the growth. RHEED patterns were recorded at an electron energy of 12 kV (RHEED 12, from STAIB Instruments).” on **Page 22, Line 379 to Page 22, Line 381 of the manuscript,**

Q21:

Line 315-42

* An explanation of the workings of the basic residual block of ResNet works is not appropriate here.

We agree with the reviewer that is not appropriate to explain the basic residual block of ResNet with a lot of details here.

We have moved the explanation of the workings of the basic residual block of ResNet 50 works from the manuscript to the supplementary information.

To make it clear, it can be found in S5 titled “The detailed structure of the ResNet 50 model” on **Page 8, Line 100 to Page 11, Line 144** and Fig. S7 of **the supplementary information.**

Q22:

* Important information on the machine learning model is missing from the methods section. There should be information on the software used, how the model was constructed, and which hardware was used. These are all missing.

We agree with the reviewer that it is important to include more details on the machine learning model.

We have provided a detailed description of the hardware configuration of the computer we are using, and the software environment and the model construction have also been elaborated in details.

To make it clear, the hardware configuration can be found in “Our model is deployed on a Windows 10 computer system with an AMD R9 7950X CPU, 64GB of memory, an NVIDIA 3090 graphics card, and a 2TB solid-state drive.” on **Page 22, Line 387** to **Page 22, Line 388** of the manuscript. “We use a USB 3.0 interface to connect the camera to the computer.” on **Page 22, Line 390** of the manuscript.

The software environment and the model construction can be found in “The model development occurred in the Jupyter Notebook environment on the Ubuntu system, utilizing Python version 3.9. We installed PyTorch based on CUDA 11.8 in this environment to leverage graphics card operations. The original video data was rapidly converted into the NumPy arrays format using code with random data augmentation within the Ubuntu system. Subsequently, the processed data was stored on our computer’s hard disk.” on **Page 22, Line 83558** to **Page 23, Line 399** of the manuscript.

With the above changes and responses, we believe we have successfully addressed all the concerns raised by the Reviewers.

We thank you for your kind consideration.

Yours Sincerely,

Prof. Chao Zhao

Key Laboratory of Semiconductor Materials Science,

Institute of Semiconductors, Chinese Academy of Sciences

Beijing 100083,

China

REVIEWER COMMENTS

Reviewer #1 (Remarks to the Author):

The authors have presented a very (!) detailed responses to all points raised by the referees. As emphasised in my last report, I like the result very much as a MBE grower myself (at least, partly). I eager to reproduce the technique in this manuscript in our experiments and to see how does it perform in other quantum dot systems. From the revisions, I see that many of the technical details are given now, allowing the peer researches to test/reproduce their works. Therefore, I have no further comment but just a full recommendation to publication.

Reviewer #2 (Remarks to the Author):

I find this article a very useful advance in the automation and analysis of QD systems grown by MBE. The authors have thoroughly answered my original concerns and modified the manuscript to address my questions. In general, this is the application of commercially successful machine learning (ML) algorithms to epitaxy. While minimal modifications of the machine learning process was performed, very little ML modifications were needed. Some reviewers may feel this implies less novelty. I disagree. The novelty is in applying the ML algorithms to successfully train a network and use that network for feedback control. I have worked on this exact same problem and find the authors results remarkable - certainly better than my own work. Knowing what details have to be circumvented (many now addressed in the manuscript after the authors changes), I have deep respect for what the authors have accomplished. I see this manuscript as the framework for a wider application of machine learning in epitaxy control.

Reviewer #3 (Remarks to the Author):

There are still some aspects in the response that I find questionable and not entirely convincing. Below are a few points of concern based on the authors' feedback:

1. Did the authors conduct the experiment again to produce Fig. R4? In the original Fig. S8, the pie chart appeared at the sequence before 1600. However, it is now shown at the sequence of around 900. Why does it happen although the raw data look the same?
2. The author claimed this in the reviewer's response: "In high-density growth experiments, the shutter remains open after the formation of QDs for a period". However, in low-density growth experiments, the closing time of the In shutter and the QD formation time almost coincide. Does this imply that QDs require a certain growth period before the In shutter is closed? In the Fig. 4g, the QDs formation time and the In shutter closing time is at the same time, does it mean that this process does not require the growing of QDs?
3. Based on my observations, the authors have retained the original graph data in Fig. S9. However, I am curious as to why the percentage of each ingredient in the pie chart has changed. Specifically, in Fig.S9, the pie chart at the sequence of around 1000 has moved to the sequence of around 600 in Fig. R5. Additionally, how did the 'high density'percentage increase in the sequence of 1000 in Fig. R5 compared to Fig. S9?
4. In the Fig. 1, the discrimination in term of QDs density is vague. In the REED images, there is not much difference between the middle and high density. However, while the highest density value is $1.601E+11$ in the 'high label', and the lowest density value is $1.25E+10$ in the 'middle label', the numerical difference between the two labels is significant. Despite this, the distinction in the REED images themselves appears minimal.
5. The authors replied that "further improvement of model accuracy becomes not obvious when selecting more than 8 images" and "It is evident that incorporating data preprocessing during the training process significantly reduces the data processing speed of the model. However, the data

processing speed of the model trained after preprocessing remains relatively constant.”. Based on the Fig. R10, could you select fewer images, such as 7, 6, or even less, to increase the data processing speed?

6. Based on my observations, I have a similar question regarding both the original and the new versions of Fig. 5g. Why was the position of the initial pie chart, which was around the sequence 1000 in the original Fig. 5g, moved to a position around the sequence 600 in the new Fig. 5g?

7. In the new Fig. 5g, why is there a possibility of a 'No' label at the sequence around 1000, even though the pie chart shows 100% 'Yes' (red color)?

8. Why did the author remove the meanings of the blue and yellow lines in the description of the new version of Fig. 5?

Reviewer #4 (Remarks to the Author):

I thank the authors for their changes so far. I will focus on the remaining areas where I have concerns. Unfortunately, I still feel that the accuracy of the machine learning models in this work is not very impressive, and further modifications are unlikely to convince me otherwise.

Q1(A):

It does not fill me with confidence that the authors are introducing previously unreported data which makes their model look better than it initially did. Why were these not presented previously?

QDs Model:

Even with this new data, I do not think that the results of the "QDs model" presented in Figure 4f is particularly impressive. It appears to incorrectly identify QD formation 7-27% of the time before QD formation, and then incorrectly predicts "no" QD formation 45% of the times after QD formation has occurred is particularly impressive.

To my understanding a perfect model would jump from 100% "no" before formation to 100% "yes" after formation. The results presented in Figure 5ff are closer to this, but the model is still predicting formation on QDs around 30 % of the time.

Density Model:

The authors state on line 205 that an output of the density model is "zero". This does not actually appear to be true, there appears to be no instances of this category in the data files provided by the authors. If the zero label does not exist, then the model is unable to make an accurate prediction prior to QD formation, and hence the results presented in figures 4f and 5h before QD formation are largely meaningless, and we are just seeing the response of a classifier being presented with data outside of the range of its training set.

The density model incorrectly classifies the film density on ~42% of epochs and ~25% of epochs for the "low" and "high" cases respectively.

A confusing feature of figure 5h is that in the cases where the model incorrectly labels the film after QD formation, it preferentially labels it as "low" density rather than medium, despite the RHEED signal being qualitatively more similar to the low density film.

Data representation:

I do not like the figures which use dots to indicate the outputs of the classifier at different timepoints (i.e. 4f,4g,5g,5h...etc). They are difficult to interpret. I requested that the authors provide a running average instead, however the plots they provided in Response Q16-(C) - Fig R20 were not what I had in mind. I'm not sure what the number on the y axis is. What I was expecting the authors to produce was a plot of the average fraction of "no"/"yes" or "low"/"middle"/"high" predicted by the model over a given window.

I also do not like the use of pie charts to demonstrate the how the model improves with time. While the authors do now provide the ranges covered by the pie charts in the main text, it should be discernable from the figure itself. It would be better if the authors actually indicated the ranges covered by each chart directly on the figure itself. A better solution however would be the running

average plot that I suggested in my previous review, and would also avoid selecting ranges that make the model look better/worse than it actually performed

Since the data is available, I generated this figure from the data in Figures 4f and 4g as an example of what I had intended the authors to produce. I used a 40-point averaging window, so the last data point is the same as the final pie chart in both figures. I also added a smoothed curve to each data series to make it easier to read.

[see attached figures]

It is apparent from this plot that there is a sudden increase in the likelihood of "yes" being predicted after QD formation. However, there is no discernable change in the density model after QD formation occurs.

Finally, why are there different pie charts in the revised Figure 4f in the manuscript and in Figure R14? Surely these should be the same, the authors should check which is the correct version of the figure.

Figure 2a/R18:

The figure is much better, but "Input" and "Output" are incorrectly spelled. Also the text of "QDs model" and "density model" is cropped.

Q11:

I did not request that the authors cite this paper. They should not feel obliged to include it.

Dear Reviewers,

We thank the Reviewers for the prompt reply and considering our work. We deeply appreciate for the critical and constructive review of the manuscript. Reviewers 1 and 2 have emphasized again the significance of the results presented in the manuscript, and we appreciate their remark. Reviewers 3 and 4 also provide very helpful comments. The suggestions have been fully addressed by incorporating additional required information, thus substantially improved the manuscript. All required changes have been addressed and highlighted in **RED** in the revised manuscript.

Our point-by-point response are included in the following for the convenience of the reviewers:

Reviewer 1:

The authors have presented a very (!) detailed responses to all points raised by the referees. As emphasised in my last report, I like the result very much as a MBE grower myself (at least, partly). I eager to reproduce the technique in this manuscript in our experiments and to see how does it perform in other quantum dot systems. From the revisions, I see that many of the technical details are given now, allowing the peer researches to test/reproduce their works. Therefore, I have no further comment but just a full recommendation to publication.

We really appreciate the comments from Reviewer 1. As mentioned in the last response letter, the methodology introduced in our work can also apply to other quantum dot systems. We wish more MBE growers will adopt our method and reproduce this technique in their experiments.

Reviewer 2:

I find this article a very useful advance in the automation and analysis of QD systems grown by MBE. The authors have thoroughly answered my original concerns and modified the manuscript to address my questions. In general, this is the application of commercially successful machine learning (ML) algorithms to epitaxy. While minimal modifications of the machine learning process was performed, very little ML modifications were needed. Some reviewers may feel this implies less novelty. I disagree. The novelty is in applying the ML algorithms to successfully train a network and use that network for feedback control. I have worked on this exact same problem and find the authors results remarkable - certainly better than my own work. Knowing what details have to be circumvented (many now addressed in the manuscript after the authors changes), I have deep respect for what the authors have accomplished. I see this manuscript as the framework for a wider application of machine learning in epitaxy control.

We thank the reviewer for appreciating our work. We indeed apply the ML algorithms to successfully train a network and use that network for feedback control. We wish to the application of machine learning will help the whole community who work on epitaxy and also other semiconductor process.

Reviewer 3:

Q1: Did the authors conduct the experiment again to produce Fig. R4? In the original Fig. S8, the pie chart appeared at the sequence before 1600. However, it is now shown at the sequence of around 900. Why does it happen although the raw data look the same?

We have fully addressed the comment.

We did not conduct the experiment again to produce Fig. R4; instead, it originated from the same raw data but a distinct data representation of the original Fig. S8.

As mentioned in our last “Response to Referees Letter”, we have improved the data representation. To effectively depict data trends, the statistical range of each pie chart data

in Fig. R4 has been adjusted compared to Fig. S8. The statistical analysis conducted on the initial 200 sequences is intended to verify whether the initial growth conditions can produce QDs with the predetermined target density. The subsequent analysis, spanning from the 200th sequence to 40 sequences before QD formation, serves as a comparative examination against the initial 200 sequences. This stage helps ascertain whether the material growth conditions have been progressively adjusted to better align with the desired growth conditions. The 40 sequences before QD formation are analyzed to demonstrate the QD model's potential to detect the initiation of QD formation. Finally, an analysis of the 40 sequences after QD formation is conducted to affirm the successful formation of QDs and to portray the ultimate outcome and density change trend of the QDs.

So, in Fig. R4, the pie chart near the 900th sequence included probability statistics of the output data of the model on sequences ranging from 200th to 40 sequences before QD formation, leading to the difference compared to the original Fig. S8.

Q2: The author claimed this in the reviewer's response: "In high-density growth experiments, the shutter remains open after the formation of QDs for a period". However, in low-density growth experiments, the closing time of the In shutter and the QD formation time almost coincide. Does this imply that QDs require a certain growth period before the In shutter is closed? In the Fig. 4g, the QDs formation time and the In shutter closing time is at the same time, does it mean that this process does not require the growing of QDs?

We have fully addressed the comment.

QD growth is a phased development process, particularly from the initial formation of QDs to the attainment of high-density QDs, which requires a significant number of atoms supplied. During the initial stages of InAs QD formation, the distribution of InAs atoms on the substrate surface is relatively scattered, and the distance between QDs is relatively wide, corresponding to low density. Closing the shutter too early in this phase results in an insufficient supply of InAs atoms, preventing the continued growth of existing QDs or the formation of new ones. Consequently, this limitation restricts the increase in both the

density and size of QDs, making it impossible to achieve high-density QDs (<https://doi.org/10.1088/0953-8984/19/22/225006>)(<https://doi.org/10.1063/1.116580>).

Therefore, high-density QDs require an additional growth period before the In shutter is closed.

The shutter closure requires the two conditions, QD formation and QD density, match the preset target. For the original Fig. 4g, the sequence when the shutter is closed indicates that the QDs have already grown, and their density has reached the preset low-density target.

To make it clear, we have already cited the two papers in the previous manuscript.

Q3: Based on my observations, the authors have retained the original graph data in Fig. S9. However, I am curious as to why the percentage of each ingredient in the pie chart has changed. Specifically, in Fig.S9, the pie chart at the sequence of around 1000 has moved to the sequence of around 600 in Fig. R5. Additionally, how did the ‘high density’ percentage increase in the sequence of 1000 in Fig. R5 compared to Fig. S9?

We have fully addressed the comment.

As mentioned in our last “Response to Referees Letter”, we have improved the data representation. To effectively depict data trends, the statistical range of each pie chart data in Fig. R5 has been adjusted compared to Fig. S9. Additionally, in high-density growth experiments, the shutter remains open after the formation of QDs for a period. Therefore, we added two additional pie charts, one before and one after the shutter is closed, to further illustrate the results of the density model. So, we are not simply moving the pie chart from about the 1000th sequence to about the 600th sequence.

Therefore, the percentage in the “high density” category in the sequence of 1000 in Fig. R5 is 20%, whereas it is 9% in Fig. S9.

Q4: In the Fig. 1, the discrimination in term of QDs density is vague. In the REED images, there is not much difference between the middle and high density. However, while the highest density value is 1.601E+11 in the ‘high label’, and the lowest density value is

1.25E+10 in the 'middle label', the numerical difference between the two labels is significant. Despite this, the distinction in the REED images themselves appears minimal.

We have fully addressed the comment.

In the original Fig. 1, the QD density is divided from low to high into the following categories: no QD, less than $1 \times 10^{10} \text{ cm}^{-2}$, from $1 \times 10^{10} \text{ cm}^{-2}$ to $4 \times 10^{10} \text{ cm}^{-2}$, more than $4 \times 10^{10} \text{ cm}^{-2}$, and corresponds to “zero”, “low”, “middle”, “high” labels, respectively.

Although the difference between Fig. 1i and 1j may appear small at first glance, upon closer observation, Fig. 1j exhibits a significantly higher brightness than Fig. 1i. Furthermore, the manuscript only presents one RHEED image, whereas in the actual material growth process, the grower observes multiple images. By comparing these images, it is easier to identify differences in changes in brightness. Additionally, the spot shape in Fig. 1i resembles a Reuleaux triangle, while the shape in Fig. 1j is more elliptical. This subtle change is due to an increase in the QDs density, resulting in a larger diffraction volume in the material and a larger area of overlapping light waves and a stronger interference effect, increasing the brightness of the diffraction spot. (Patella F, Arciprete F, Fanfoni M, et al. Reflection high energy electron diffraction observation of surface mass transport at the two-to three-dimensional growth transition of InAs on GaAs (001). *Applied Physics Letters*, 2005, 87(25)) (Gunasekera M, Freundlich A. Real time during growth metrology and assessment of kinetics of epitaxial quantum dots by RHEED//2012 38th IEEE Photovoltaic Specialists Conference. IEEE, 2012: 001794-001797) (Shimomura K, Shirasaka T, Tex D M, et al. RHEED transients during InAs quantum dot growth by MBE. *Journal of Vacuum Science & Technology B*, 2012, 30(2))

Even though the differences are subtle, they can still be identified through careful observation.

To make it clear, we cited the mentioned papers in the revised manuscript.

Q5: The authors replied that “further improvement of model accuracy becomes not obvious when selecting more than 8 images” and “It is evident that incorporating data preprocessing

during the training process significantly reduces the data processing speed of the model. However, the data processing speed of the model trained after preprocessing remains relatively constant.”. Based on the Fig. R10, could you select fewer images, such as 7, 6, or even less, to increase the data processing speed?

We thank the reviewer for pointing it out.

The selection of fewer images as input to the model can have a certain impact on processing speed. However, richer input information can also enhance the model’s capabilities. Therefore, we recommend striking a balance between data processing speed and performance. In our research that demands real-time feedback control, it is essential to consider both factors. Our selection meets the requirement for feedback control by using 8 images as inputs, resulting in the model outputting at least 1 result per second.

Q6: Based on my observations, I have a similar question regarding both the original and the new versions of Fig. 5g. Why was the position of the initial pie chart, which was around the sequence 1000 in the original Fig. 5g, moved to a position around the sequence 600 in the new Fig. 5g?

We apologize for the misunderstanding.

As mentioned, to effectively depict data trends, the statistical range of each pie chart data in the new versions of Figure 5g has been adjusted compared to the original Figure 5g. Additionally, in high-density growth experiments, the shutter remains open after the formation of QDs for a period. Therefore, we added two additional pie charts, one before and one after the shutter is closed, to further illustrate the results of the density model. So, we are not simply moving the pie chart from around the sequence 1000 in the original Fig. 5g to around the sequence 600 in the new Fig. 5g.

Q7: In the new Fig. 5g, why is there a possibility of a ‘No’ label at the sequence around 1000, even though the pie chart shows 100% ‘Yes’ (red color)?

We have fully addressed the comment.

In the new Fig. 5g, the statistical range of the pie chart located at the sequence around 1000 extends from the final 40 sequence before the blue line to the blue line, which is 1019th sequence to 1058th sequence. However, the last model output “No” label is at 1018th sequence. Therefore, the last “No” label is not within the statistical range of the pie chart, resulting in the pie chart showing 100% “Yes”.

Q 8. Why did the author remove the meanings of the blue and yellow lines in the description of the new version of Fig. 5?

We have fully addressed the comment.

To avoid repeated explanation, we have revised the manuscript and defined the blue and yellow lines at lines 254-255 in the manuscript before describing the experimental data, rather than explaining them in each figure caption.

Reviewer 4:

Q1: It does not fill me with confidence that the authors are introducing previously unreported data which makes their model look better than it initially did. Why were these not presented previously?

We thank the reviewer for raising this question.

We would like to clarify that these previously unreported data were included from suggestions provided by Reviewer 2 and Reviewer 3. After careful consideration of the comments from reviewers, and to improve the comprehensiveness of our study, we have included previously unreported data to provide a more complete and robust analysis.

Q2: QDs Model:

Even with this new data, I do not think that the results of the "QDs model" presented in

Figure 4f is particularly impressive. It appears to incorrectly identify QD formation 7-27% of the time before QD formation, and then incorrectly predicts "no" QD formation 45% of the times after QD formation has occurred is particularly impressive.

To my understanding a perfect model would jump from 100% "no" before formation to 100% "yes" after formation. The results presented in Figure 5ff are closer to this, but the model is still predicting formation on QDs around 30 % of the time.

We have fully addressed the comment.

The growth process of QDs progresses from the nucleation to the formation of QDs. (Yamaguchi K, Kaizu T, Yujobo K, et al. Uniform formation process of self-organized InAs quantum dots. *Journal of crystal growth*, 2002, 237: 1301-1306; Krzyzewski T J, Joyce P B, Bell G R, et al. Role of two-and three-dimensional surface structures in InAs-GaAs (001) quantum dot nucleation. *Physical Review B*, 2002, 66(12): 121307). The difference between the low-density QD RHEED before and after QD formation is less significant, that is whether there are small spot features in addition to the streak features. Not only are these small spots difficult to see, but they are often covered by streak features, making them difficult to detect. So, before the formation of QDs, about 7-27% of cases result in incorrect identify of QD formation. After QD formation, although extremely small spots are present, they are often obscured by streak features, leading to a 45% incorrect predicts that QDs have not formed.

QD growth is a phased development process, particularly from the initial formation of QDs to the attainment of high-density QDs, which requires a significant number of atoms supplied. During the initial stages of InAs QD formation, the distribution of InAs atoms on the substrate surface is relatively scattered, and the distance between QDs is relatively wide, corresponding to low density. Closing the shutter too early in this phase results in an insufficient supply of InAs atoms, preventing the continued growth of existing QDs or the formation of new ones. Consequently, this limitation restricts the increase in both the density and size of QDs, making it impossible to achieve high-density QDs (<https://doi.org/10.1088/0953-8984/19/22/225006>)(<https://doi.org/10.1063/1.116580>).

Therefore, high-density QDs require an additional growth period before the In shutter is closed. So, although the QDs model has determined that the QD has been formed around the 1050th sequence in Fig. 5g, the low-density QD has just been formed. Consequently, the experiment utilized approximately 30% of the sequence for the formation of high-density QDs.

In short, due to the unique nature of QD growth, achieving a model with 100% accuracy is challenging. It is important to note that our study is the first to apply ML algorithms to an epitaxial growth scenario. While we acknowledge the existing gap compared to perfect models, we believe this disparity will diminish with future study. As highlighted by Reviewer 2, the significance of our research lies in applying the ML algorithms to successfully train a network and use that network for feedback control.

Q3: Density Model:

The authors state on line 205 that an output of the density model is "zero". This does not actually appear to be true, there appears to be no instances of this category in the data files provided by the authors. If the zero label does not exist, then the model is unable to make an accurate prediction prior to QD formation, and hence the results presented in figures 4f and 5h before QD formation are largely meaningless, and we are just seeing the response of a classifier being presented with data outside of the range of its training set.

The density model incorrectly classifies the film density on ~42% of epochs and ~25% of epochs for the "low" and "high" cases respectively.

A confusing feature of figure 5h is that in the cases where the model incorrectly labels the film after QD formation, it preferentially labels it as "low" density rather than medium, despite the RHEED signal being qualitatively more similar to the low density film.

We have fully addressed the comment.

The model utilized in the study is designed to predict the density of QDs after growth. The "zero" label corresponds to a rare growth condition where QDs cannot form, which contradicts the experimental goal of achieving QD formation. Additionally, this category

has indeed appeared in the data file of Fig. 4g provided by us, but with rare appearance. We did not include them in the data plot during the data processing, as it has little impact on the overall data interpretation.

To make it clear, we included the following sentence, “In addition, since the “zero” labels appear rarely and have little impact on output results, it was not included in the figures.” on **Page 15, Line 255** to **Page 15, Line 257** of the manuscript.

The density of QDs gradually increases with deposition, however, there might be a subsequent decrease in QD density due to ripening of the QDs (Tatebayashi J, Nishioka M, Someya T, et al. Area-controlled growth of InAs quantum dots and improvement of density and size distribution. *Applied Physics Letters*, 2000, 77(21): 3382-3384; Frigeri P, Nasi L, Prezioso M, Seravalli L, Trevisi G, Gombia E, Mosca R, Germini F, Bocchi C, Franchi S. Effects of the quantum dot ripening in high-coverage InAs/GaAs nanostructures. *Journal of Applied Physics*, 2007, 102(8)). So, even if the model identifies a current QD density as low, there is still a probability that the density will increase with continued growth time, resulting in approximately 42% of “middle” and “high” labels in Fig. 4g. Similarly, when the model predicts a high density after QD formation, there is still a probability, that the density will decrease over time, resulting in approximately 25% “low” and “middle” labels in Fig. 5h.

After the formation of QDs in Figure 5h, the corresponding material growth reaches the critical thickness, and the QD density is still relatively low. Therefore, as shown in RHEED, the model output is mainly labeled as “low”. However, as the conditions become more suitable for the formation of high-density QDs, more QDs will quickly form on the surface, and RHEED images will directly change from low density to high density, with few “middle” labels appearing.

Q4: Data representation:

I do not like the figures which use dots to indicate the outputs of the classifier at different timepoints (i.e. 4f,4g,5g,5h...etc). They are difficult to interpret. I requested that the authors provide a running average instead, however the plots they provided in Response Q16-(C) -

Fig R20 were not what I had in mind. I'm not sure what the number on the y axis is. What I was expecting the authors to produce was a plot of the average fraction of "no"/"yes" or "low"/"middle"/"high" predicted by the model over a given window.

I also do not like the use of pie charts to demonstrate the how the model improves with time. While the authors do now provide the ranges covered by the pie charts in the main text, it should be discernable from the figure itself. It would be better if the authors actually indicated the ranges covered by each chart directly on the figure itself. A better solution however would be the running average plot that I suggested in my previous review, and would also avoid selecting ranges that make the model look better/worse than it actually performed

Since the data is available, I generated this figure from the data in Figures 4f and 4g as an example of what I had intended the authors to produce. I used a 40-point averaging window, so the last data point is the same as the final pie chart in both figures. I also added a smoothed curve to each data series to make it easier to read.

[see attached figures]

We really appreciate the Reviewer for pointing out the issue and suggesting the running average plot to improve the data representation.

Stimulated by the Reviewer' comment, we have adjusted the figures and modified the corresponding description in the Manuscript and Supplementary Info. Indeed, the figures are much easier to interpret. The trend now can be discernable from the figure itself.

To make it clear, we included the following sentence, "Figure 4f shows the "QDs

model” results of the experiment with the “low” label as the target. To effectively depict data trends, we have incorporated running average plots to illustrate the evolution of the model’s output results. We classified RHEED videos as “Yes” or “No” based on observations of QDs formation. At the beginning of growth, the primary output of the QDs model was “No” from the initial to the 200th sequence, and the probability of outputting “Yes” was only 7%. Even though the proportion of outputting “Yes” increases to 15%, the likelihood of the model outputting “No” remains higher from the 200th to the final 40 sequence before the blue line. When the growth process approaches the QDs formation, we observed a significant increase from 15% to over 27% of the “Yes” label. This trend continues until the shutter closes, with over 55% probability of outputting “Yes”.” on **Page 16, Line 269 to Page 16, Line 278** and Fig. 4f of **the manuscript**.

“Figure 4g shows the “density model” results of the experiment with the “low” label as the target. From the initial to the 200th sequence, the probability of output “low” is significantly lower than that of combined “middle” and “high”. However, with the real-time feedback control of growth conditions, it is evident that the probability of output “low” approaches over 50%. The growth conditions have been gradually adjusted to better suit the growth of low-density QDs. It becomes apparent from the sequence before and after the blue line that the probability of the model outputting “low” surpasses 55% and persists until the end of the growth.” on **Page 16, Line 281 to Page 16, Line 287** and Fig. 4g of **the manuscript**.

“In high-density growth experiments, the shutter remains open after the formation of QDs for a period. As depicted in Figure 5g, the QDs model predominantly outputs “No” before QDs formation, with the probability of outputting “Yes” reaching a maximum of 30%, as evidenced from the initial to around 1000th sequence. As we approach the formation of QDs, the probability of the QDs model outputting “Yes” approaches 100% around the blue line, indicating high accuracy in recognizing spot patterns after the QD formation.” on **Page 18, Line 308 to Page 18, Line 313** and Fig. 5g of **the manuscript**.

“The output results of the density model are also presented in Figure 5h. Before QDs formation, the density model rarely outputs “high”. However, as the growth process

approaches the stage of QD formation, the probability of “high” label output slowly increases to 15% around the blue line. Following the formation of QDs, the probability of the density model outputting “high” rapidly increases to over 30% around the 1100th sequence. Due to the unique nature of QD systems, closing the shutter at this point would restrict the formation of high-density QDs. As the growth process continues and approaches the moment the shutter is closed, it becomes evident that the probability of the model outputting “high” rapidly increases to over 70%.” on **Page 18, Line 316 to Page 19, Line 323** and Fig. 5h of **the manuscript**.

“To effectively depict data trends, we have incorporated running average plots to illustrate the model’s output evolution. Before QDs formation, the model predominantly outputs “No” from the initial to around 1200th sequence, as depicted in Figure S15f. However, as the sequence approaches the blue line, the likelihood of observing “Yes” significantly rises, surpassing 55% and continuing to increase to 65% at the final sequence. It indicates the model’s sensitivity in detecting the emergence of low-density QDs.” on **Page 19, Line 258 to Page 19, Line 263** and Fig. S15f of **the supplementary information**.

“From the initial to around the 600th sequence, the cumulative probability of the model outputting “middle” and “high” labels surpasses the probability of outputting “low” labels, which indicates that the initial conditions may not be suitable for the growth of low-density QDs. From around the 600th to around the 1200th sequence, we can observe a gradual increase in the probability of the density model outputting “low”. This trend continues until the final sequence, with the probability of the “low” label remaining stable at around 60%.” on **Page 19, Line 264 to Page 19, Line 270** and Fig. S15g of **the supplementary information**.

“In Figure S16f, from the initial to around the 600th sequence, there is a consistently low probability at 30% of output “Yes”. However, when approaching the QD formation at around the 800th sequence, a distinct trend emerges: the probability of the QDs model outputting “Yes” gradually increases to 72%.” on **Page 21, Line 293 to Page 21, Line 296** and Fig. S16f of **the supplementary information**.

“Furthermore, in Figure S16g, due to the initial conditions favoring high-density QD

growth, the probability of outputting a “low” label is significantly lower than the combined probability of “middle” and “high” labels. From the initial to the 200th sequence, the presence of the “low” label in the model’s output is approximately 23%. Then, the likelihood of the model outputting the “low” label increases gradually from the 200th to the 700th sequence, surpassing 33%. Finally, the density model consistently produces “low” outputs around the blue line, finally reaching 62%.” on **Page 21, Line 297** to **Page 21, Line 302** and Fig. S16g of **the supplementary information**.

Q5:

It is apparent from this plot that there is a sudden increase in the likelihood of "yes" being predicted after QD formation. However, there is no discernable change in the density model after QD formation occurs.

We have fully addressed the comment.

Indeed, there is no discernable change in the density model after QD formation occurs from the figure the reviewer generated from the data in Figure 4g. The output of the density model relies on the RHEED videos during growth. Once a low-density QD is formed, the shutter closes and there is little change in the subsequently collected RHEED videos. In comparison, during the growth process of high-density QDs, the model can quickly detect this more obvious change where the growth is going on with the shutter open, as shown in the Fig. 5h.

Q6: Finally, why are there different pie charts in the revised Figure 4f in the manuscript and in Figure R14? Surely these should be the same, the authors should check which is the correct version of the figure.

We apologize for any inconvenience caused during the review process.

After verification, we confirmed that Fig. 4f is correct.

Q7: Figure 2a/R18:

The figure is much better, but "Input" and "Output" are incorrectly spelled. Also the text of "QDs model" and "density model" is cropped.

We thank the reviewer for raising this important question.

We believe that the Reviewer was mentioning Fig. 3a instead of 2a. We have checked the spelling in the image and made slight adjustments to the text arrangement to avoid the text being cropped, as shown in Fig. R1.

Fig R1. A simplified architectural diagram of the model.

To make it clear, this is included in Figure 3a of the manuscript.

Q8: Q11:

I did not request that the authors cite this paper. They should not feel obliged to include it.

We thank the reviewer for raising this question.

We understand that the reviewer did not request that we cite this paper. However, we found it to be closely aligned with our own work, particularly in the context of the research background section. The cited paper provides valuable insights that enhance the overall coherence and depth of our study. We would like to emphasize that the inclusion of this citation is intended to provide a broader understanding of the research landscape and to acknowledge relevant contributions.

With the above changes and responses, we believe we have successfully addressed all the concerns raised by the Reviewers.

We thank you for your kind consideration.

Yours Sincerely,

Prof. Chao Zhao

Key Laboratory of Semiconductor Materials Science,

Institute of Semiconductors, Chinese Academy of Sciences

Beijing 100083,

China